# 5′ UTR length shapes alternative N-terminal protein isoforms across cancers and in rare disease

Jimmy Ly [1,2], Eric M Smith [1], Matteo Di Bernardo [1,2], Yi Fei Tao [1,2], Elizabeth M Black [1], Ekaterina Khalizeva [1,2] & Iain M Cheeseman [1,2]✉

## Abstract

The 5′ untranslated region (5′ UTR) of an mRNA is classically viewed as a regulatory region that controls the amount of protein production, but not the resulting protein sequence. Here, we demonstrate that 5′ UTR length plays a direct role in alternative N-terminal protein isoform production by controlling start codon selection. We find that very short 5′ UTRs enhance leaky ribosome scanning, thereby promoting the production of truncated alternative N-terminal protein isoforms. We also show that endogenous changes in 5′ UTR length due to alternative transcription initiation can tune the relative abundance of alternative N-terminal isoforms from the same gene. In addition, we identify mutations in rare genetic diseases that alter 5′ UTR length, including a deletion in the VHL 5′ UTR in von Hippel–Lindau disease that shifts translation toward the shorter VHLp19 isoform. Together, our results implicate 5′ UTR length as a determinant of alternative N-terminal isoform production and reveal an underappreciated mechanism by which noncoding changes can reshape the proteome.

**Subject Categories** Cancer; Translation & Protein Quality

## Introduction

For decades, eukaryotic mRNAs were generally assumed to encode a single protein product. This view has been revised by growing evidence that many transcripts initiate translation at multiple, in-frame alternative start codons, producing distinct N-terminal protein isoforms from a single mRNA (Andreev et al, 2022; Eisenberg et al, 2020; Ingolia et al, 2011; Kearse and Wilusz, 2017; Morrison and Rissland, 2025; Wright et al, 2022). These alternative N-terminal isoforms can differ from their annotated counterparts in their interacting partners (Muller et al, 2010; Sun et al, 2021; Tsang and Cheeseman, 2023), subcellular localization (Higdon et al, 2024; Lama-Diaz and Blanco, 2024; Lee et al, 2024; Ly et al, 2025), stability (Gawron et al, 2016; Saris et al, 1991; Trulley et al,

2019), and other fundamental aspects of protein biology (Fukushima et al, 2012; Thomas et al, 2008). Alternative translation initiation substantially expands proteomic diversity beyond current annotations and contributes broadly to cellular physiology and disease. Defining the mechanisms that govern the production of alternative N-terminal isoforms is therefore essential for understanding this previously hidden layer of gene expression.

Eukaryotic mRNAs are typically organized into three functional regions: the protein-coding open reading frame (ORF), an upstream 5′ untranslated region (5′ UTR), and a downstream 3′ untranslated region (3′ UTR), with the 5′ UTR playing key roles in regulating translational efficiency (Leppek et al, 2018; Reimao-Pinto et al, 2025; Strayer et al, 2024). Numerous cis-regulatory features within the 5′ UTR can influence start codon selection (Brito Querido et al, 2024; Dever et al, 2023; Gu et al, 2021; Kozak, 2002), including the Kozak context for the start codon (Ivanov et al, 2010; Kozak, 1987; Ly et al, 2024b; Pisarev et al, 2006; Zhou et al, 2020), the use of non-AUG initiation codons (Andreev et al, 2022; Eisenberg et al, 2020; Kearse and Wilusz, 2017; Ly et al, 2024b; Grosely et al, 2025), RNA secondary structure (Kozak, 1990; Lee et al, 2024; Wang et al, 2022; Xiang et al, 2023), and upstream open reading frames (uORFs) (Dever et al, 2023; Fernandez et al, 2024; Schleich et al, 2014). In contrast, the length of the 5′ UTR itself has not traditionally been considered a major determinant of the corresponding protein products that are produced, including alternative N-terminal isoform production. Prior work by Marilyn Kozak demonstrated that shortening the 5′ UTR of a reporter mRNA to below 30 nucleotides (nt) promotes leaky ribosomal scanning (Fujimoto et al, 2022; Kozak, 1991). In addition, recent structural studies support a model in which the ribosome engages the 5′ end of the mRNA in a manner that occludes a 40–50 nt region proximal to the cap from undergoing robust initiation (Brito Querido et al, 2020). Consequently, start codons positioned very close to the 5′ cap may be inefficiently recognized or bypassed completely. Although these findings suggest 5′ UTR length could act as an important determinant of start codon selection in vitro, the extent to which endogenous transcripts exploit this mechanism to produce alternative N-terminal isoforms remains largely unexplored.

Here, we use ribosome profiling (Ingolia et al, 2011; Ly et al, 2024b) and Cap Analysis of Gene Expression (CAGE)-sequencing (Shiraki et al, 2003) to identify endogenous mRNAs that undergo

[1]Whitehead Institute for Biomedical Research, Cambridge, MA, USA. [2]Department of Biology, Massachusetts Institute of Technology, Cambridge, MA, USA.
✉E-mail: icheese@wi.mit.edu

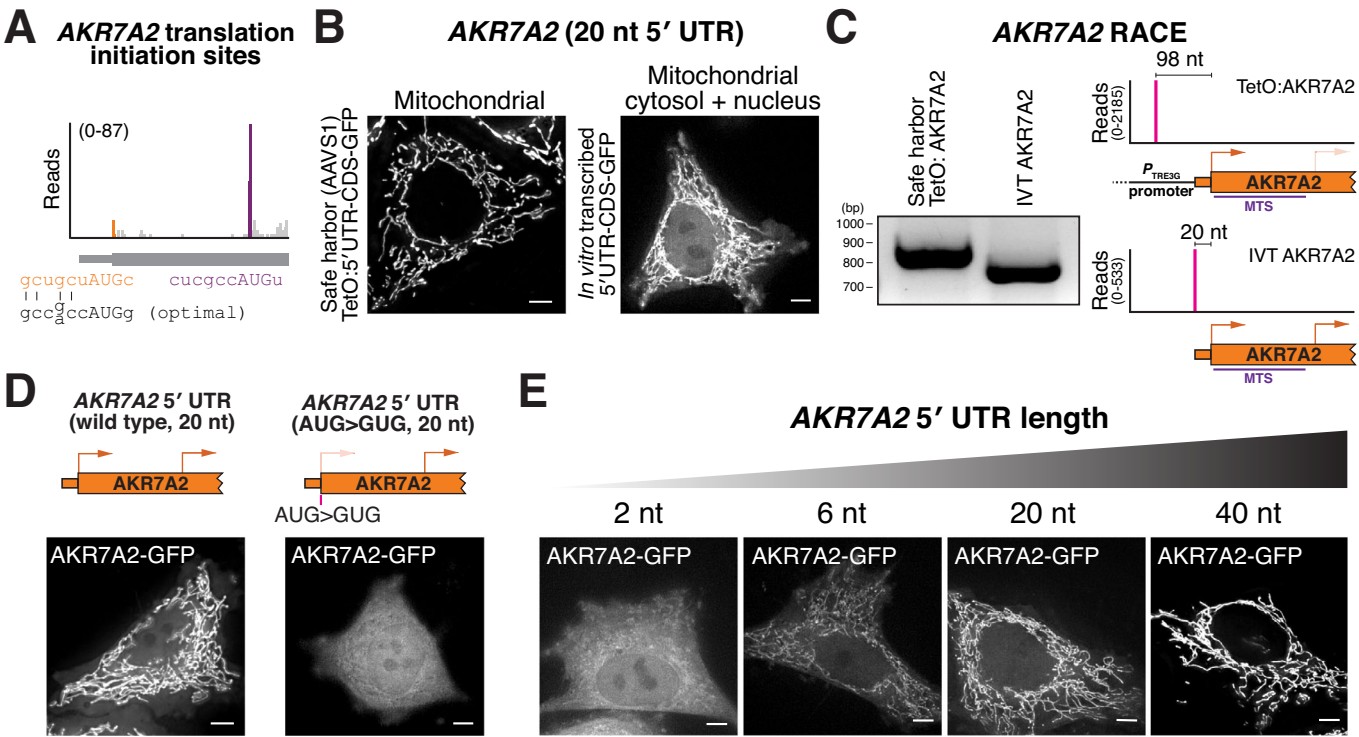

**Figure 1.  5′ UTR length regulates AKR7A2 alternative N-terminal isoform selection.**

(**A**) Read distribution from translation initiation site profiling around the *AKR7A2* alternative start codons. The orange peak represents reads at the annotated start codon, whereas the purple reads represent the alternative N-terminally truncated start site. Optimal Kozak context from (Kozak, 1987). (**B**) Live-cell imaging of AKR7A2-GFP produced from the dox-inducible promoter or in vitro-transcribed mRNA. (**C**) Left, 5′ RACE of endogenous AKR7A2 from transgenic AKR7A2 expressed under a dox-inducible promoter and in vitro-transcribed *AKR7A2* mRNA. Right, quantification of 5′ RACE reads by sequencing. (**D**) Left, live-cell imaging of AKR7A2 after transfecting mRNA containing the AKR7A2 5′ UTR – CDS – GFP. Right, localization of AKR7A2, where the annotated start codon was mutated to GUG. (**E**) Live-cell imaging of cells transfected with in vitro-transcribed *AKR7A2* 5′ UTR – CDS – GFP with the indicated 5′ UTR lengths. Scale bar represents 5 µm. Source data are available online for this figure.

short-5′-UTR-driven leaky scanning to produce functional alternative N-terminal isoforms. We further show that alternative transcription initiation can modulate 5′ UTR length, thereby altering protein isoform production without changing the coding sequence. Finally, we identify pathogenic patient alleles that alter 5′ UTR length and consequently shift alternative N-terminal isoform ratios, highlighting the physiological importance of 5′ UTR lengths.

# Results

## 5′ UTR length regulates alternative N-terminal isoform selection for AKR7A2

Our recent work using start site ribosome profiling (Ly et al, 2024b) identified hundreds of human genes that encode dual protein products with distinct subcellular localizations (Ly et al, 2025). Importantly, a subset of these mRNAs exhibits alternative start codon usage that cannot be readily explained solely by poor Kozak context-mediated leaky scanning. One such example is *AKR7A2* (Li et al, 2012), which contains two in-frame translation start sites as revealed by start site ribosome profiling (Fig. 1A). Translation from the annotated upstream AUG generates a mitochondrial protein,

whereas initiation at a downstream AUG produces a cytosolic/nuclear isoform (Fig. EV1A; Ly et al, 2025). To determine how these AKR7A2 isoforms are decoded, we expressed an AKR7A2 construct containing its native 5′ UTR and complete coding sequence fused to GFP. When expressed from a dox-inducible promoter ($P_{TRE3G}$; Das et al, 2016; Loew et al, 2010), this 5′ UTR-AKR7A2-GFP construct produced exclusively mitochondrial localization, indicating strict initiation at the first AUG (Fig. 1B). Although the upstream AUG has a moderate strength Kozak context (Fig. 1A), this result suggests that Kozak strength alone is insufficient to drive leaky scanning for this transcript to allow downstream initiation. In contrast, transfection of an in vitro-transcribed *AKR7A2* 5′ UTR–CDS–GFP mRNA resulted in clear dual localization to mitochondria, cytosol, and nucleus (Fig. 1B), consistent with usage of both start codons and the behavior of endogenous *AKR7A2*.

To understand the difference in AKR7A2 localization produced from the dox-inducible promoter vs. the in vitro-transcribed mRNA, we considered the nature of these constructs. Most promoters used for ectopic expression in human cells will append additional nucleotides upstream of the cloned 5′ UTR to the transgenic mRNA (Glasner et al, 2025; Kim et al, 1990; Niwa et al, 1991). For example, the dox-inducible promoter ($P_{TRE3G}$) used for

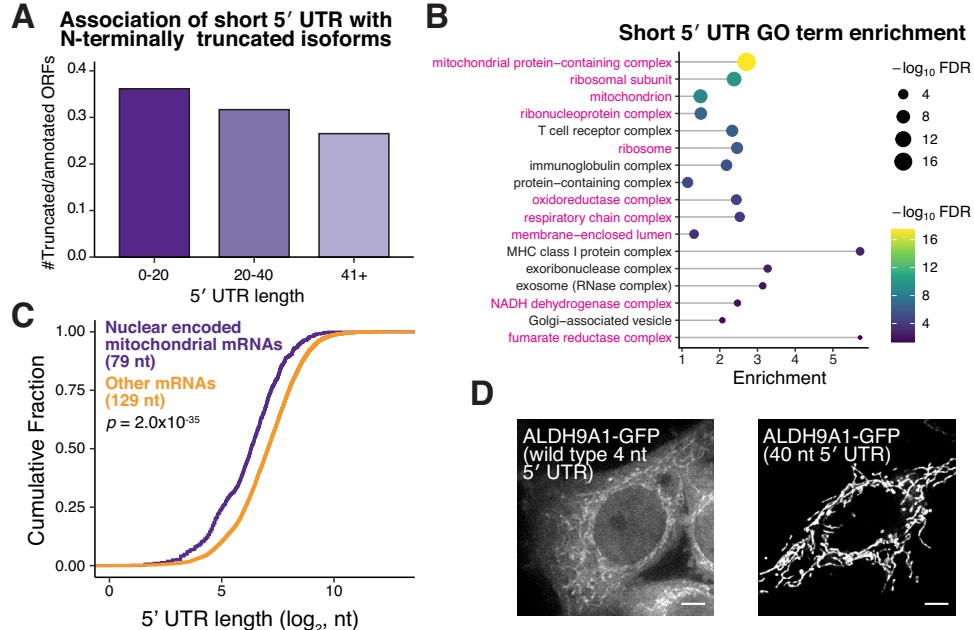

**Figure 2. Short 5′ UTR regulation affects many mitochondrial genes.**

(A) Bar plot showing the relative proportion of N-terminal truncations to annotated ORFs for genes in different 5′ UTR length bins, indicating that mRNAs with shorter 5′ UTRs have more N-terminal truncations. (B) GO term analysis of human genes with short 5′ UTRs. Mitochondria-associated GO terms are highlighted in magenta. (C) Cumulative distribution fraction plots showing the 5′ UTR lengths of nuclear-encoded mitochondrial genes compared to all other nuclear-encoded genes. Statistic represents results from the Wilcoxon rank-sum test. (D) Live-cell imaging of cells transfected with in vitro-transcribed ALDH9A1 5′ UTR – CDS – GFP with the wild-type (4 nt) and elongated (40 nt) 5′ UTR. Scale bar represents 5 μm. Source data are available online for this figure.

our experiments will add ~78 nucleotides to the cloned mRNA sequence (Fig. 1C). Thus, the dox-inducible mRNA has a longer 5′ UTR than the in vitro-transcribed *AKR7A2* construct.

The native *AKR7A2* 5′ UTR is unusually short, only 20 nt, whereas the average 5′ UTR length in humans is around 200 nucleotides (Leppek et al, 2018). To test whether 5′ UTR length underlies these observed differences in alternative N-terminal isoform production, we transfected cells with in vitro-transcribed *AKR7A2* mRNAs containing defined 5′ UTRs lengths and a C-terminal GFP tag. The native 20-nt *AKR7A2* 5′ UTR supported production of both mitochondrial and cytosolic/nuclear isoforms, whereas mutation of the first start codon (AUG to GUG) abolished mitochondrial localization, confirming that relative localization reflects initiation at the two start codons (Fig. 1D). Extending the 5′ UTR to 40 nt using the Xenopus β-globin sequence suppressed downstream initiation and yielded exclusively mitochondrial protein localization (Fig. 1E). Conversely, further shortening the 5′ UTR to 6 or 2 nt shifted initiation toward the downstream start site, favoring cytosolic/nuclear localization (Fig. 1E).

The alternative start sites and predicted dual localization for AKR7A2 is highly conserved across mammals. Notably, mouse *AKR7A2* also has an unusually short 11-nt 5′ UTR and produces dual-localized protein isoforms in a manner that depends on 5′ UTR length (Fig. EV1B–D). However, 5′ UTR length is not universally conserved as chimpanzee and gorilla *AKR7A2* transcripts have substantially longer 5′ UTRs (67 and 127 nt, respectively) (Fig. EV1B). Overall, these results demonstrate that the short 5′ UTR length of *AKR7A2* in a subset of organisms strongly influences the production of its differentially localized alternative N-terminal isoforms.

## Nuclear-encoded mitochondrial mRNAs are enriched for short 5′ UTRs

Based on our analysis of AKR7A2, we hypothesized that other mRNAs with short 5′ UTRs would be more likely to produce N-terminally truncated protein isoforms. To evaluate whether short 5′ UTR lengths are correlated with the production of alternative translational isoforms on a global level, we conducted a computational analysis of the relationship between 5′ UTR length (based on annotated mRNAs; (Dyer et al, 2025)) and downstream translation initiation (based on start site ribosome profiling; (Ly et al, 2024b)). We found that mRNAs with short 5′ UTRs (≤ 40 nt) produced a larger frequency of alternative N-terminally truncated isoforms compared to those mRNAs with longer 5′ UTRs (> 40 nt) (Figs. 2A and EV2A,B; Dataset EV1). Amongst diverse biological processes, we found that nuclear-encoded genes that produce mitochondrial-localized proteins, such as AKR7A2, are enriched for short 5′ UTRs (Figs. 2B,C and EV2C–E).

Based on our analysis, we identified at least 75 human genes with short 5′ UTRs that produce more than one translational isoform based on start site ribosome profiling (Dataset EV1). To test these, we focused on ALDH9A1 (Chen et al, 2024; Jung et al, 2025), which has an annotated 5′ UTR length of 4 nt (Fig. EV2D). Indeed, using mRNA transfection, the annotated *ALDH9A1* mRNA produced dual-localized mitochondrial and cytosolic isoforms (Fig. 2D). In contrast, increasing the 5′ UTR length to 40 nt biased translation towards the mitochondrial isoform, whereas reducing the length biased towards the production of the cytosolic isoform

(Figs. 2D and EV3A,B). Notably, *ALDH9A1* also has a poor Kozak context and strengthening the context also promotes translation of the mitochondrial isoform (Fig. EV3C,D). This suggests that *ALDH9A1* has two sequence elements—a weak Kozak context and an unusually short 5′ UTR—that promote usage of the downstream start codon and the cytosolic isoform.

We next examined the conservation of the short *ALDH9A1* 5′ UTR length. Analysis of *ALDH9A1* 5′ UTR lengths across mammals based on the Ensembl database (Dyer et al, 2025) revealed that the short 5′ UTR is not highly conserved (Fig. EV3E). Despite this, orangutan *ALDH9A1* mRNA, which contains a 40-nt 5′ UTR, still produced dual-localized ALDH9A1 protein isoforms (Fig. EV3F,G). Analysis of the first AUG start codon for orangutan *ALDH9A1* revealed a weak Kozak context, which may promote the production of dual-localized ALDH9A1 isoforms (Fig. EV3C). These results suggest that the conservation of the ALDH9A1 poor Kozak context may be enough to maintain the production of dual-localized ALDH9A1 translational isoforms.

Overall, these data suggest that 5′ UTR length regulation can act as a broad paradigm to affect the translation of alternate protein isoforms, contributing to diverse biological processes and expanding the functional human proteome, with a particular role in mitochondrial function.

## Alternative transcription regulates start codon selection of AKR7A2 and complement factor D (*CFD*)

Given the importance of 5′ UTR length in dictating protein isoform selection, we next considered mechanisms that could modulate 5′ UTR length to influence start codon selection. One such mechanism is alternative transcription initiation, which can generate mRNAs with distinct 5′ UTRs. Alternative transcription initiation in budding yeast has been shown to influence the production of alternative N-terminal protein isoforms by changing the 5′ end of the mRNA to include or exclude start codons (Higdon et al, 2024). To identify analogous cases in human cells, we performed Cap Analysis of Gene Expression (CAGE; (Shiraki et al, 2003)) in HeLa cells to identify alternative transcription start sites (Fig. 3A). In total, we identified 81,079 transcription start sites, with 86% (7714/8314) of genes displaying more than one start site (Dataset EV2). Of genes with more than one transcription start site, our analysis revealed 8468 alternative transcription initiation sites that are downstream of the annotated start codon and are thus predicted to cause the production of a truncated protein isoform (Dataset EV2). For example, AKR7A2 has a downstream promoter that excludes the annotated start codon in the mRNA such that this transcript fails to translate the mitochondrial isoform (Fig. EV4A). As a result, transcripts initiated from this alternative promoter can only produce the truncated nuclear/cytosolic isoform. Thus, AKR7A2 uses multiple mechanisms to generate the alternative truncated protein isoform, including short 5′ UTR–driven leaky scanning from the annotated mRNA isoform (ENST00000235835) as well as alternative promoter usage.

A second compelling example of alternate promoters affecting translation start site usage is the Complement Factor D (CFD) gene (Sekine et al, 2023), which contains two conserved translation initiation sites (Figs. 3B and EV4B). These sites are predicted to generate alternative N-terminal isoforms with distinct subcellular localizations (Fig. EV4C,D). To evaluate this, we tested the

localization of each isoform in HeLa cells. Indeed, translation from the annotated start codon produced a Golgi-localized protein (Fig. 3C), consistent with CFD's role as a secreted complement factor (Sekine et al, 2023). In contrast, initiation at the downstream site produced a truncated isoform that localized to the cytosol and nucleus (Fig. 3C). Notably, we found that expression of the annotated *CFD* mRNA produces only the Golgi-localized isoform (Fig. 3D,E), indicating that this transcript does not undergo leaky ribosomal scanning. Instead, 5′ CAGE analysis revealed an alternative downstream transcription start site (Fig. 3D) that removes the annotated start codon and enforces production of the truncated CFD isoform (Fig. 3E).

The alternative start codon and differential localization for CFD isoforms are highly conserved across vertebrates (Fig. EV4E). However, a functional role for the intracellular CFD isoform has yet to be described. To test this, we next investigated the role of this truncated CFD isoform. Based on immunoprecipitations coupled with quantitative mass spectrometry, the annotated, secreted CFD isoform interacted with only two ER/Golgi-associated proteins, consistent with its localization to the secretory pathway (Fig. 3F). In contrast, the truncated isoform showed a broad interaction network, including numerous proteins involved in genome organization and transcription (Fig. 3G,H; Dataset EV3). CFD is not essential for viability in cultured human cells (DepMap; (Tsherniak et al, 2017)), limiting functional analysis of the intracellular isoform in cultured cell lines. Nevertheless, ClinVar (Landrum et al, 2018) reports mutations (ClinVar accessions: VCV001511411.6 and VCV001494493.5) that are classified as M40V or M40I missense mutations in the annotated CFD isoform, but are also predicted to disrupt the translation of alternative N-terminally truncated intracellular CFD. Whether the pathogenicity is mediated by the alternative start codon mutation or the missense effect on the annotated CFD isoform remains to be further characterized. This raises the possibility that the alternative CFD isoform may perform a physiologically important intracellular function, with differential promoter usage affecting the relative production of these isoforms across conditions.

Overall, these data highlight the importance of promoter selection in regulating the production of alternative N-terminal isoforms.

## Alternative transcription can control 5′ UTR length to regulate isoform selection of ALDH9A1 and GUK1

The examples above illustrate how substantial differences in alternative transcription start sites can remove upstream start codons, redirecting translation to downstream sites. Importantly, our analysis of *AKR7A2* and *ALDH9A1* (Figs. 1 and 2) highlighted that small changes in 5′ UTR length alone—without altering coding sequences—can also tune protein isoform production. To test whether alternative transcription can change the production of alternative N-terminal isoforms by modulating 5′ UTR length, we analyzed our CAGE-seq data to identify alternative promoters that strictly change 5′ UTR length (Dataset EV2). For example, the annotated *ALDH9A1* transcript has a 4 nt 5′ UTR, but alternative transcription initiation extends the 5′ UTR to 28 nt (Fig. 4A). Notably, we found that the longer 28 nt 5′ UTR isoform increased initiation at the first start codon, elevating the relative levels of the mitochondrial isoform (Fig. 4B,C).

Reciprocally, we identified alternative transcription start sites that shorten the 5′ UTR relative to the annotated transcript. GUK1

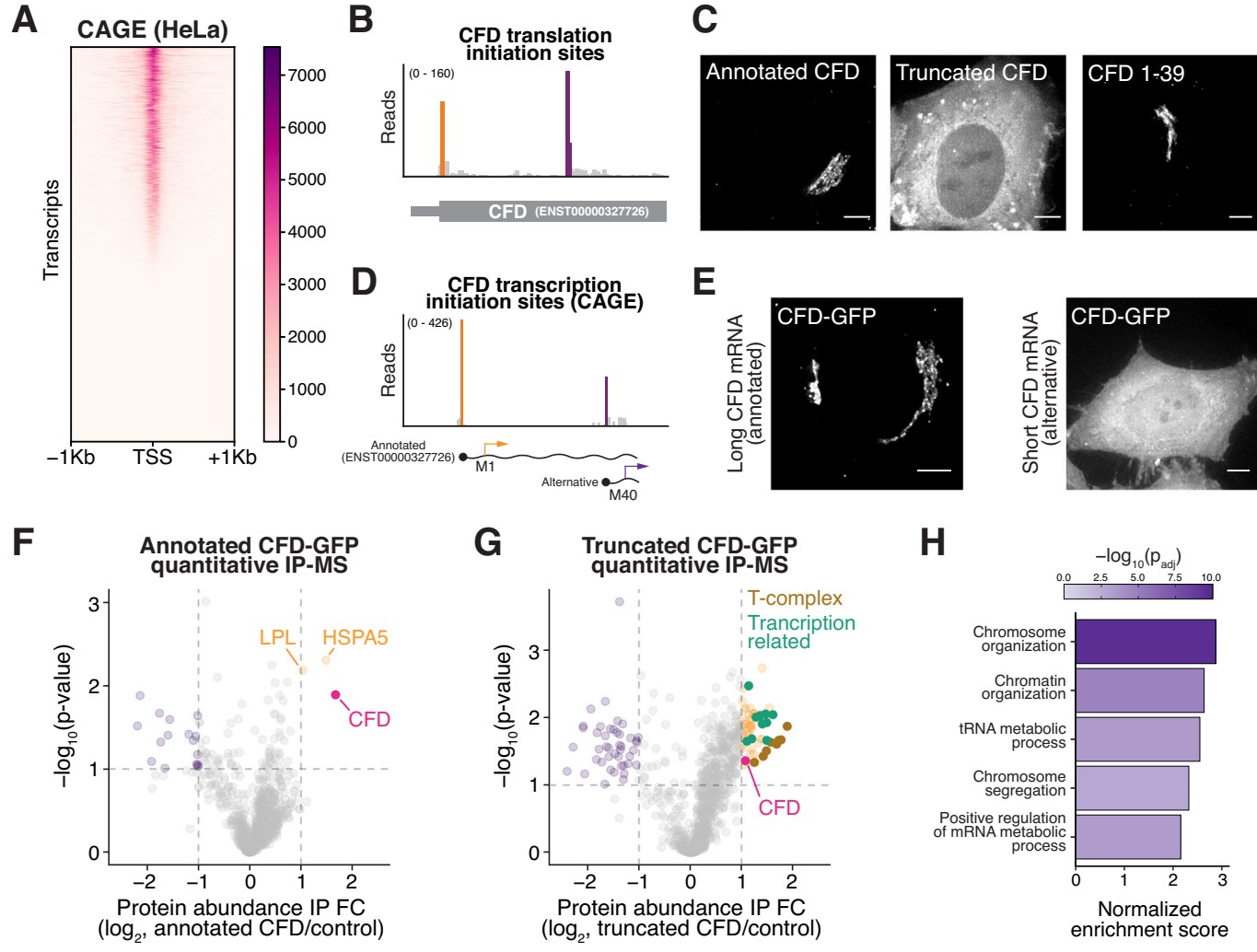

**Figure 3. Alternative transcription of CFD produces an intracellular alternative N-terminal isoform.**

(A) Heatmap of CAGE-Seq data from HeLa cells. Scale bar represents scaled read counts. (B) Read distribution trace from translation initiation site profiling around the CFD alternative start codons. (C) Live-cell imaging of annotated CFD, Truncated CFD isoforms and the N-terminus of annotated CFD tagged with a C-terminal GFP. Images are not scaled equally. (D) Read distribution traces from CAGE-Seq data around the CFD transcription start sites. (E) Live-cell imaging of HeLa cells expressing the annotated CFD mRNA isoform (left) or the alternative transcript isoform (right). Images are not scaled equally. Scale bar represents 5 μm. (F) Volcano plot of quantitative IP-MS analysis of annotated CFD. (G) Volcano plot of quantitative IP-MS analysis of truncated CFD. $n = 2$ biological replicates. The following selected genes were highlighted as transcription-related: ZNF609, CDK5RAP2, RAD21, NFAT5, EMSY, SUPT16H, ATRX, SMC1A, KDM1A, NCAPD2. One-way ANOVA was used for the statistics. (H) Gene set enrichment analysis for truncated CFD interactors. Permutation test was used for statistics and Benjamini–Hochberg FDR correction was applied for multiple hypothesis testing. Source data are available online for this figure.

(Hidalgo-Gutierrez et al, 2024; Schneider et al, 2025), which contains two translation initiation sites (Fig. EV5A), has an annotated 5′ UTR of 28 nt. However, usage of an alternative transcription start site generates an mRNA isoform with a 2 nt 5′ UTR without changing the coding sequence (Fig. 4D). Because the alternate translation initiation sites in *GUK1* do not produce differentially localized protein isoforms (Fig. EV5B), we generated a reporter construct to assess the usage of these two start sites (Fig. 4E). This reporter construct contains two in frame AUG start codons with an intervening nuclear export signal followed by a C-terminal GFP (Fig. 4E). Usage of the different start codons will produce protein products of different sizes such that the relative

ratio of the proteins will reflect the relative extent of leaky ribosome scanning. By comparing the amount of leaky scanning for the annotated *GUK1* 5′ UTR (28 nt) relative to the alternative *GUK1* 5′ UTR (2 nt), we observed enhanced leaky ribosome scanning for the 2 nt 5′ UTR (Figs. 4F and EV5C). Therefore, despite encoding identical open reading frames, endogenous transcripts with differing 5′ UTR lengths will produce distinct ratios of GUK1 protein isoforms. These data highlight the importance of considering alternative transcripts with different 5′ UTR lengths when interpreting the protein products generated from a given gene.

Our prior work identified cell-type-specific alternative transcription of KNSTRN/SKAP, which results in the production of a

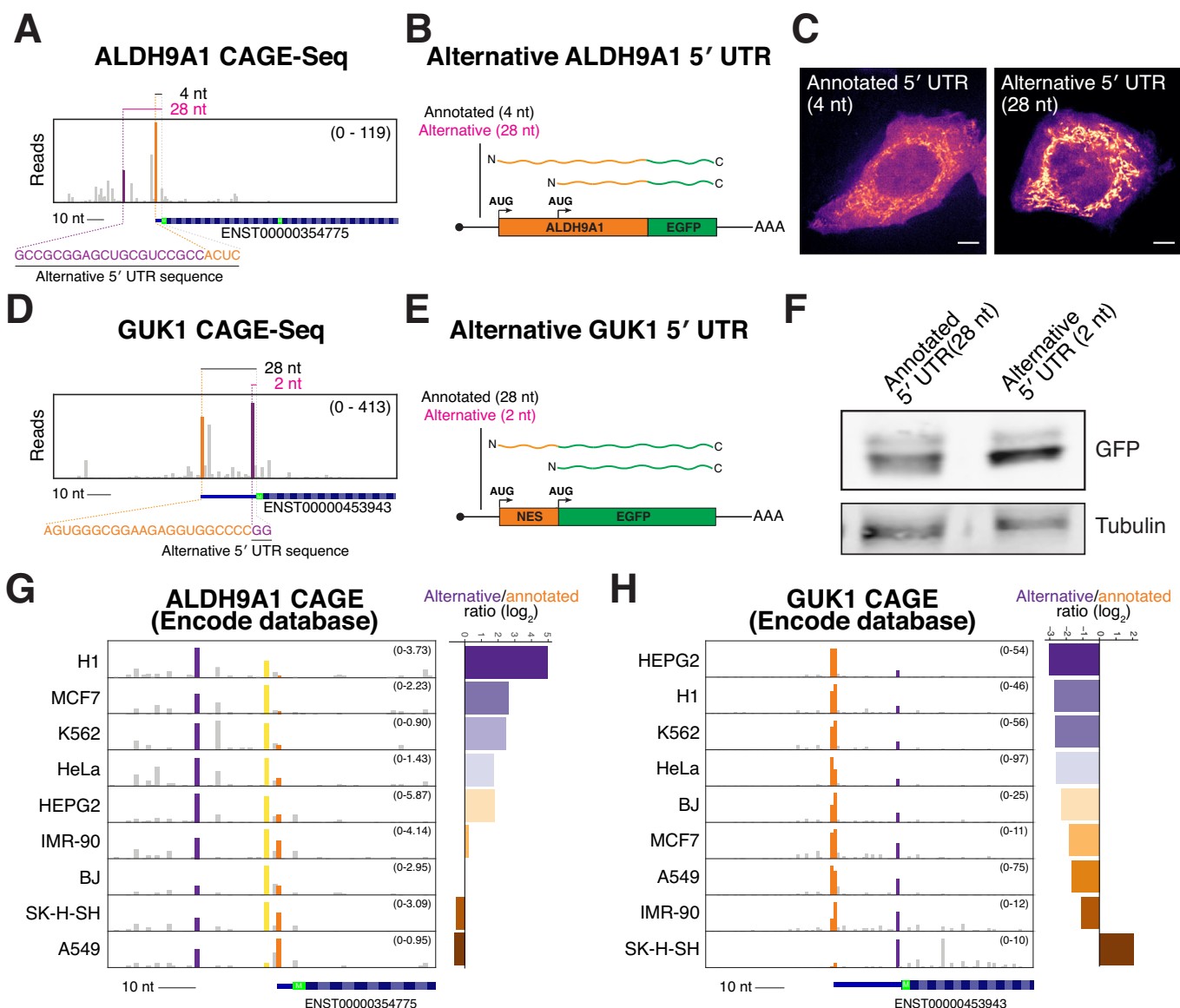

**Figure 4. Alternative transcription initiation modulates 5′ UTR length to regulate alternative N-terminal isoform ratios.**

(A) CAGE-Seq analysis of ALDH9A1 transcription start sites. Orange reads represent the annotated transcription start site, and purple reads represent the alternative transcription start site, resulting in a longer 5′ UTR. (B) Schematic of in vitro-transcribed ALDH9A1 reporter mRNA with annotated or alternative 5′ UTRs. The relative ratio of mitochondrial compared to nuclear/cytosolic isoform represents the relative start codon usage. (C) Live-cell imaging for wild-type (4 nt) and alternative (28 nt) *ALDH9A1* reporters. Scale bar indicates 5 μm. (D) Same as (A) except for GUK1. (E) Schematic of in vitro-transcribed *GUK1* 5′ UTR reporters. The GUK1 5′ UTRs were appended to a nuclear export signal flanked with AUG start codons followed by a GFP. (F) Western blot showing the relative ratios of the protein products produced by the first start codon (larger protein) vs the second start codon (shorter protein). (G) CAGE-seq trace for ALDH9A1 transcription start sites across 9 cell lines from the ENCODE consortium (Consortium, 2012). The orange read indicates reads at the annotated transcription start site, and the purple indicates an alternative longer 5′ UTR isoform. The yellow reads indicate another alternative start site, but only extending it to 7 nt. The quantification on the right indicates the relative number of alternative promoter reads normalized to the annotated start site. (H) Same as (G) except with GUK1. Source data are available online for this figure.

distinct, testes-specific N-terminal isoform (Kern et al, 2016). Thus, changes in 5′ UTR length could result in differing tissue or context-specific production of alternate protein isoforms. To test this, we examined whether alternative transcription initiation of ALDH9A1 and GUK1 varies across biological contexts, thereby modulating alternative N-terminal isoform production solely by changing 5′ UTR length. Based on analysis of CAGE-seq datasets generated by the ENCODE consortium (Consortium, 2012), we identified

diverse promoter usage across diverse cancer cell lines. For both ALDH9A1 and GUK1, transcription start site usage varies markedly across a panel of 9 cancer cell lines (Fig. 4G,H), with the ratios of longer to shorter 5′ UTR transcripts spanning 48-fold for ALDH9A1 and 31-fold for GUK1. These results indicate that alternative transcription initiation of ALDH9A1 and GUK1 differs across cell types and may function to tune N-terminal isoform abundance in a context-specific manner.

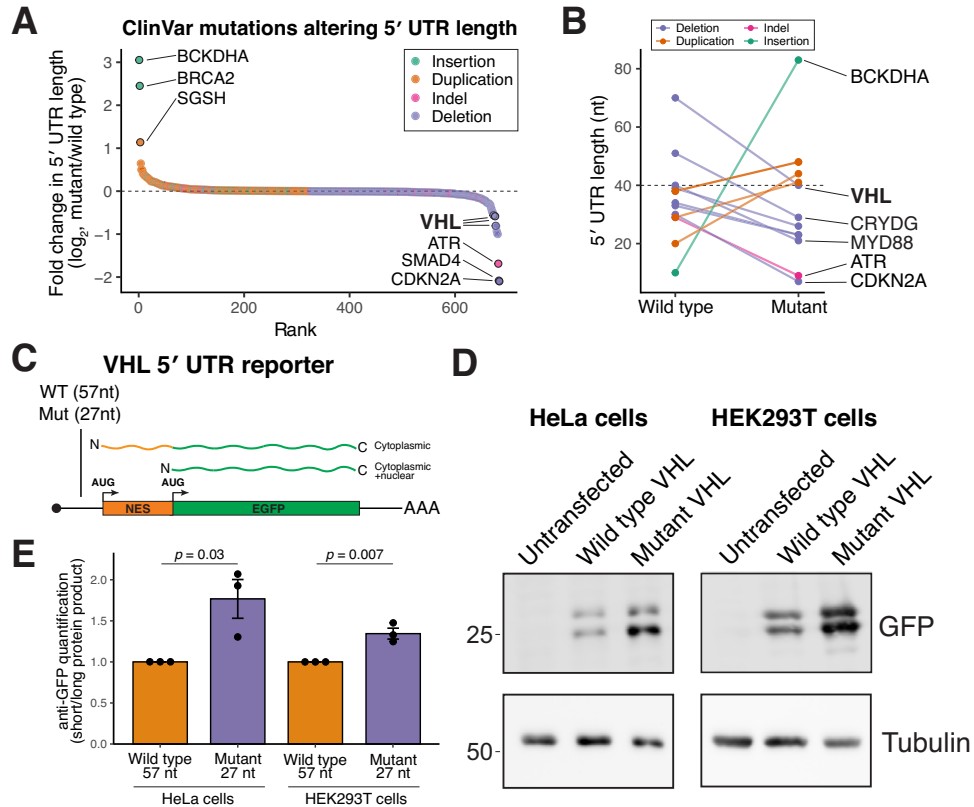

**Figure 5. 5′ UTR disease mutations can alter alternative N-terminal isoform ratios.**

(A) Waterfall plot showing the fold change in 5′ UTR length induced by ClinVar mutations. (B) Subset of ClinVar 5′ UTR mutations that alter length within the 5′ UTR range of 40 nt. (C) Schematic of VHL 5′ UTR reporter. The wild-type and mutant VHL 5′ UTR are driving the expression of a nuclear export signal flanked by two start codons, followed by GFP. The first start codon produces a NES-GFP that should be enriched in the cytosol, whereas the second start produces just GFP which should localize to both the nucleus and cytosol. (D) Representative western blot from HeLa (left) or HEK293 (right) cells transfected with the indicated in vitro-transcribed VHL reporters. (E) Quantification of western blots in (D). Error bar indicates standard error of the mean, $n = 3$ biological replicates, and statistics indicate Student's $T$ test. Source data are available online for this figure.

## Pathogenic 5′ UTR mutations regulate alternative start codon selection through 5′ UTR length control

The analysis of rare human disease typically focuses on coding mutations (Whiffin et al, 2020; Wieder et al, 2025; Wieder et al, 2024). However, our analysis of 5′ UTR lengths suggests that noncoding changes could also alter relative isoform production. Thus, we considered whether there are pathological mutations that alter 5′ UTR length and start codon selection. To test this, we mined the ClinVar database (Landrum et al, 2018) for rare disease alleles that are predicted to alter 5′ UTR length. In total, we identified 988 alleles across 519 genes that are predicted to change 5′ UTR length (Fig. 5A; Dataset EV4). Most variants were small insertions, duplications, or deletions of one or two nucleotides. However, we identified several variants that are predicted to cause substantial alterations in 5′ UTR length, ranging from −411 nt to +889 nt (Fig. 5A).

We next focused on variants predicted to have the strongest impact on start codon selection by affecting transcripts with very short 5′ UTRs (< 40 nt). We identified multiple ClinVar mutations that are predicted to increase or decrease short UTR by at least 10 nt (Fig. 5B). We identified 5 mutations that lengthen short 5′ UTRs,

which we predict would suppress downstream leaky ribosome scanning, and 5 mutations that shorten these already short 5′ UTRs, which are predicted to enhance leaky scanning (Dataset EV4). In addition, we identified cases in which pathogenic deletions convert long 5′ UTRs into short (< 40 nt) UTRs, a change expected to promote leaky ribosome scanning. Using these criteria, we identified CRYGD, MYD88, and VHL as candidate genes in which disease-associated variants may induce or enhance leaky ribosome scanning resulting in a N-terminally truncated protein isoform.

To test the impact of these 5′ UTR changes on alternative N-terminal isoform selection, we focused on VHL. VHL is a tumor suppressor that functions as part of an E3 ubiquitin ligase complex that targets hypoxia-inducible factor (HIF) for ubiquitination and degradation (Gossage et al, 2015). Mutations in VHL cause von Hippel–Lindau disease, a hereditary cancer syndrome (Maher et al, 2011). VHL is alternatively translated through leaky ribosome scanning to produce ~30 kDa and ~19 kDa isoforms (Blankenship et al, 1999; Iliopoulos et al, 1998), with unique functions (Falconieri et al, 2020; Hergovich et al, 2003; Hergovich et al, 2006; Minervini et al, 2015). Notably, mutations that affect the ratio of alternative VHL isoforms differentially

contribute to disease and pathology (Bartels et al, 2015; Frew et al, 2013).

The annotated *VHL* mRNA contains a 5′ UTR of 70 nt. However, empirical mapping of transcription start sites by CAGE revealed that the predominant *VHL* 5′ UTR in HeLa cells is shorter, at 57 nt (Fig. EV5D). We therefore used the GFP reporter described above (Fig. 4E) to assess start codon selection driven by either the wild-type *VHL* 5′ UTR (57 nt) or a pathogenic 30 nt 5′ UTR deletion mutant (VCV002928001.4) (Figs. 5C and EV5D). The 5′ UTR deletion enhanced leaky ribosome scanning and is therefore predicted to increase production of the shorter VHL p19 isoform (Figs. 5D,E and EV5E,F). Together, these results suggest that disease-associated 5′ UTR mutations can directly regulate alternative start codon usage, altering N-terminal isoform output and highlighting the importance of evaluating 5′ UTR variants beyond their canonical regulatory roles. As noncoding variants are typically under-reported in rare disease databases (Ellingford et al, 2022; Wieder et al, 2025), the prevalence of such alleles may be an underrepresentation of potential genetic impacts across human disease.

# Discussion

Defining the cis-regulatory features that control the relative production of alternative N-terminal isoforms in mammals has important relevance to core cellular function and disease. Here, we demonstrate that endogenous 5′ UTR length is an important determinant that tunes the relative usage of alternative start codons. For example, both *AKR7A2* and *ALDH9A1* mRNAs undergo leaky ribosomal scanning to generate distinct mitochondrial and nuclear/cytosolic N-terminal isoforms. Notably, these transcripts have exceptionally short 5′ UTRs (20 nt for *AKR7A2* and 4 nt for *ALDH9A1*) and extending the 5′ UTR length suppresses leaky scanning in a graded manner, promoting the production of their longer, mitochondrial isoforms (Figs. 1 and 2). We note that AKR7A1 and ALDH9A1 lack TISU elements—sequences that promote usage of the cap-proximal start codon in mRNAs with short 5′ UTRs (Elfakess et al, 2011; Sinvani et al, 2015). Whether other sequences promote or repress leaky scanning in these mRNAs remains to be determined.

Although our data are consistent with increased leaky scanning past the first start codon in mRNAs with short 5′ UTRs, additional mechanisms may also contribute. Recent structural studies suggest that mRNA loading into the 43S pre-initiation complex can occur through a "ribosome slotting" mode, in which the ribosome engages the mRNA in a manner that creates a blind spot at the 5′ end and prevents sampling of start sites in this region (Brito Querido et al, 2024). The alternative start codon usage observed for short 5′ UTR transcripts could therefore reflect in vivo ribosome slotting rather than, or in addition to, classical leaky scanning. Whether there are physiological conditions or sequence elements that bias towards slotting versus scanning represents an important future direction.

The 5′ UTR length of *AKR7A2* and *ALDH9A1* appears to vary across mammals. This may reflect the possibility that these transcripts may not be alternatively translated in all mammals. However, differences in 5′ leader length do not necessarily preclude truncated isoform production, as additional regulatory mechanisms may compensate. For example, *ALDH9A1* contains an evolutiona-rily conserved weak Kozak context that likely promotes leaky

scanning independent of 5′ UTR length (Fig. EV3C–E). Similarly, despite having a long 5′ UTR and strong Kozak context, chimpanzee *AKR7A2* is expressed as two mRNA isoforms (XM_001163068.7 and XM_063785489.1) capable of generating dual-localized protein products. Finally, errors in transcript annotation across species may also contribute to the apparent divergence in *ALDH9A1* and *AKR7A2* 5′ UTR architecture.

The importance of 5′ UTR length in regulating alternative isoform production has practical implications for transgenic expression systems, as commonly used promoters often append additional sequences to the 5′ end of transcripts. For example, the TRE3G promoter adds 78 nt to the 5′ UTR (Fig. 1C), abolishing leaky scanning of *AKR7A2* and *ALDH9A1*. Thus, studies of alternative start codon usage must carefully consider reporter design, since plasmid- or transgene-based systems can inadver-tently alter 5′ UTR architecture or introduce confounding effects on transcription and splicing (Loughran et al, 2025). In such cases, in vitro-transcribed mRNA reporters with defined 5′ ends may provide a more accurate approach for assessing start codon selection.

Based on existing genome annotations, there are 2899 genes with at least one mRNA isoform containing a short 5′ UTR (Dataset EV1). However, because not all of these transcripts are expressed in HeLa cells—the cell line used for our ribosome profiling analysis (Ly et al, 2024b)—we cannot directly assess whether each of these genes also use alternative translation start sites. Nonetheless, given their short 5′ UTR lengths, the potential for alternative translation initiation should be considered when analyzing these genes.

We further show that alternative promoter usage can alter 5′ UTR length, and that even modest changes are sufficient to shift the relative production of N-terminal protein isoforms. For example, the annotated *GUK1* transcript contains a 28-nt 5′ UTR, whereas an alternative transcription start site generates an mRNA with a 2-nt 5′ UTR, promoting leaky ribosome scanning and altered start codon selection. Thus, endogenous changes in promoter usage can tune start codon selection efficiency and reshape protein isoform output. Notably, this alternative *GUK1* transcript is absent from current genome annotations, under-scoring how incomplete transcript models obscure proteomic diversity.

Finally, by analyzing the ClinVar database (Landrum et al, 2018), we identified disease-associated alleles predicted to change 5′ UTR length (Fig. 5). Strikingly, 10 of these alleles fall within a short-UTR regime—either occurring in transcripts with 5′ UTRs shorter than 40 nt or truncating longer 5′ UTRs to below this 40 nt threshold—where leaky scanning is predicted to be induced. Among these, we identified pathogenic variants in the VHL 5′ UTR linked to Von Hippel–Lindau syndrome that markedly shorten the 5′ UTR (Fig. 5). We show that these mutations promote leaky ribosome scanning, thereby shifting translation toward a truncated VHL isoform. Together, these findings establish altered 5′ UTR length as a previously underappreciated disease mechanism, demonstrating that noncoding mutations can drive pathology not by altering mRNA stability or overall translational efficiency, but also by rewiring start codon selection. As such, 5′ UTR mutations represent a critical and often overlooked class of pathogenic variants that should be systematically considered in human genetic disease.

# Methods

**Reagents and tools table**

| Reagent/resource | Reference or source | Identifier or catalog number |
| --- | --- | --- |
| **Experimental models** | | |
| HeLa cells | ATCC | CVCL_0030 |
| HEK293T cells | ATCC | CVCL_0063 |
| **Recombinant DNA** | | |
| TRE3G: Annotated human AKR7A2 5′ UTR-CDS-EGFP | Ly et al, 2025 | pJL597 |
| TRE3G: Truncated AKR7A2-EGFP | Ly et al, 2025 | pJL598 |
| TRE3G: Annotated mouse AKR7A2 5′ UTR-CDS-EGFP | This study | pJL872 |
| TRE3G: Annotated human ALDH9A1 CDS-EGFP | Ly et al, 2025 | pJL806 |
| TRE3G: Orangutan ALDH9A1 5′ UTR-CDS-EGFP | This study | pJL1046 |
| TRE3G: CFD 5′ UTR-CDS-EGFP | This study | pJL569 |
| TRE3G: truncated CFD-EGFP | This study | pJL570 |
| TRE3G: CFD$^{1-39}$-EGFP | This study | pJL669 |
| NES-GFP dual start codon reporter | This study | pJL939 |
| **Antibodies** | | |
| Anti-GFP | Roche | 11814460001 (AB_390913) |
| Anti-α-tubulin | Abcam | ab52866 (AB_869989) |
| IRDye 680RD Goat anti-Rabbit | LI-COR | 926-68071 (AB_10956166) |
| IRDye 680RD Goat anti-Mouse | LI-COR | 926-68070 (AB_10956588) |
| IRDye 800CW Goat anti-Rabbit | LI-COR | 926-32211 (AB_621843) |
| IRDye 800CW Goat anti-Mouse | LI-COR | 926-32210 (AB_621842) |
| **Oligonucleotides and other sequence-based reagents** | | |
| Template-switching oligo | Ly et al, 2025 | oJL1137 |
| AKR7A2 RACE oligo | This study | oJL1466 |
| AKR7A2 nested RACE oligo | This study | oJL1467 |
| **Chemicals, enzymes, and other reagents** | | |
| MycoAlert Mycoplasma Detection Kit | Lonza | LT07-418 |
| Maxima First Strand cDNA Synthesis Kit | Thermo Fisher Scientific | K1641 |
| Lipofectamine 2000 | Thermo Fisher Scientific | 11668027 |
| Lipofectamine MessengerMAX | Thermo Fisher Scientific | LMRNA008 |
| HiScribe T7 ARCA mRNA Kit (with tailing) | NEB | E2060 |

| Reagent/resource | Reference or source | Identifier or catalog number |
| --- | --- | --- |
| Monarch RNA Cleanup Kit | NEB | T2040 |
| EconoSpin All-in-1 Mini Spin Columns | Epoch Life Science | 1920-250 |
| Template Switching RT Enzyme Mix | NEB | M0466 |
| TRI Reagent | Invitrogen | AM9738 |
| S-Trap micro columns | Protifi | C002-MICRO-0080PK |
| Trypsin | Promega | V5111 |
| TMT10plex kit | Thermo Fisher Scientific | 90111 |
| Pierce High-pH peptide fractionation kit | Thermo Fisher Scientific | 84868 |
| **Software** | | |
| CellProfiler | Carpenter et al, 2006 | https://cellprofiler.org |
| Kallisto | Bray et al, 2016 | https://pachterlab.github.io/kallisto |
| fgsea (R package) | Bioconductor | |
| STAR aligner | Dobin et al, 2013 | https://github.com/alexdobin/STAR |
| bedtools | Quinlan and Hall, 2010 | |
| deepTools | Ramirez et al, 2014 | |
| IGV | Thorvaldsdottir et al, 2013 | https://software.broadinstitute.org/software/igv/ |
| Proteome Discoverer | Thermo Fisher Scientific | |
| 5′ UTR ClinVar analysis | This study | https://github.com/cheeseman-lab/5-utr-clinvar-mutations |

## Cell culture

HeLa (CVCL_0030) and HEK293T (CVCL_0063) cells were maintained in DMEM supplemented with 10% heat-inactivated fetal bovine serum, 2 mM L-glutamine, and 100 U/mL penicillin–streptomycin at 37 °C in a humidified incubator with 5% $CO_2$. Cell lines were routinely tested for mycoplasma contamination using the Lonza MycoAlert Mycoplasma Detection Kit (LT07-418).

## Molecular biology

All cDNA used in this study, with the exception of constructs derived from non-human genes (mouse AKR7A2 and Orangutan ALDH9A1), was amplified from HeLa cell cDNA. The mouse AKR7A2 and orangutan ALDH9A1 cDNA were synthesized by Twist Bioscience. To increase 5′ UTR length, the *Xenopus* β-globin 5′ UTR (5′-GAATACAAGCTTCTTGTTCTTTTTGCAGAAGCT-CAGAATAAACGCTCAACTTTGGCAGATCTGAAC-3′) was appended upstream of the 5′ UTR of interest to achieve the

desired length. When only a portion of the β-globin 5′ UTR was required to reach a specific length, sequences were taken from the 3′ end of the β-globin UTR. To prepare HeLa cell cDNA, whole cell HeLa cell RNA was reverse transcribed using the Maxima First Strand cDNA Synthesis Kit (K1641) according to the manufacturer's protocol.

All transgenic constructs were inserted downstream of a doxycycline-inducible promoter (Loew et al, 2010). The donor plasmid encoded a puromycin resistance cassette, the reverse doxycycline-controlled transactivator, and the transgene of interest, flanked by homology arms targeting the AAVS1 safe-harbor locus (Smith et al, 2008). HeLa cells were co-transfected with 500 ng of the donor plasmid and 500 ng of pX330-AAVS1 sgRNA using Lipofectamine 2000. Forty-eight hours after transfection, cells were subjected to selection with 0.45 μg/mL puromycin for a minimum of 3 days.

For the AKR7A2, ALDH9A1, in vitro-transcribed mRNA reporters 5′ UTR and coding sequence of interest was cloned directly downstream of the T7 promoter and upstream of a C-terminal GFP. For the GUK1 and VHL 5′ UTR reporters, the 5′ UTR was incorporated into the oligonucleotide and used to amplify a plasmid containing an AUG start codon followed by the protein kinase inhibitor (PKI) nuclear export signal (LALKLAGLDI), an AgeI restriction site, a 15-amino-acid glycine–serine linker, a second AUG start codon, and a C-terminal GFP. The glycine–serine linker was included to increase the size difference between products for western blot analysis.

## In vitro transcription, capping, polyadenylation, and purification

Plasmids containing the reporter gene with a C-terminal GFP tag were amplified via PCR with an oligo with the T7 promoter (5′-TAATACGACTCACTATAGGG-3') to generate templates for in vitro transcription. The resulting PCR products were digested with DpnI to remove plasmid template, purified using EconoSpin™ All-in-1 Mini Spin Columns (Epoch Life Science, 1920-250), and eluted with RNase-free water.

For in vitro transcription, the purified PCR product was transcribed using the HiScribe T7 ARCA mRNA Kit with tailing (NEB, E2060) according to the manufacturer's instructions. RNA purification steps were performed using the Monarch Spin RNA Cleanup Kit (NEB, T2040) as recommended by the manufacturer. Purified mRNA was eluted in nuclease-free water, quantified using a Nanodrop, aliquoted, flash-frozen with liquid nitrogen, and stored at −80 °C.

## mRNA transfections

Cells were seeded in 12-well plates and grown to ~80% confluency prior to transfection. mRNA transfections were carried out using Lipofectamine MessengerMAX (Thermo Fisher, LMRNA008) according to the manufacturer's protocol, using 1 μg of mRNA and 2 μL of reagent per well. Eight hours after transfection, cells were gently dissociated by incubation with PBS supplemented with 5 mM EDTA for 5 min at 37 °C. Cells were then replated onto glass-bottom 12-well imaging plates at ~60% confluency and allowed to recover overnight.

## Live-cell imaging

For dox-inducible transgenes, cells were induced for 16 h with 1 μg/mL doxycycline prior to imaging. Cells were incubated with Hoechst dye (0.1 μg/mL) for 30 min prior to imaging. Imaging was performed in a temperature-controlled chamber using a DeltaVision Ultra imaging system (Cytiva) equipped with a 60×/1.42 NA objective. Z-stack images spanning 8 μm were acquired with a step size of 0.2 μm.

## Image quantification

CellProfiler (Carpenter et al, 2006) was used to quantify the nuclear-to-cytoplasmic ratio of GFP expressed from VHL 5′ UTR reporter constructs. Nuclei were first segmented based on DNA staining using a global Otsu thresholding strategy with two classes and a minimum and maximum object size of 100–300 pixels. Cell boundaries were then defined from each nucleus using the GFP signal and the propagation method, with global Otsu thresholding and three classes.

Integrated GFP intensity was measured separately for the nuclear and cytoplasmic compartments. To restrict the analysis to transfected cells expressing the reporter, a minimum integrated GFP intensity threshold of 500 was applied to both nuclear and cytoplasmic signals.

## RNA extractions

Cells were detached using PBS supplemented with 5 mM EDTA, pelleted in DMEM, and rinsed once with PBS. The resulting pellet was lysed in 400 μL TRI reagent (Invitrogen, AM9738). Chloroform (120 μL) was added, and samples were mixed vigorously followed by centrifugation at 21,000×g for 15 min at 4 °C. The upper aqueous phase was transferred to a new tube, extracted again with an equal volume of chloroform, and centrifuged at 21,000×g for 1 min at 4 °C. For RACE, the RNA was precipitated by supplementing the aqueous phase with 300 mM NaCl and 30 μg GlycoBlue (AM9516), followed by the addition of an equal volume of isopropanol and incubation at −20 °C overnight. For CAGE, the RNA was precipitated with LiCl (2.5 M final) at −20 °C overnight. RNA was pelleted by centrifugation at 21,000×g for 30 min at 4 °C, washed once with 70% ethanol, and resuspended in RNase-free water. RNA concentration was determined using a Nanodrop.

## 5′ rapid amplification of cDNA end (RACE) and sequencing

5′ RACE (Yeku and Frohman, 2011) was carried out using the Template Switching RT Enzyme Mix (NEB, M0466) following the supplier's recommendations. For each reaction, 1 μg of total RNA was used as input for first-strand cDNA synthesis. Gene-specific reverse-transcription primers were first annealed to RNA by heating for 5 min at 70 °C. Reverse transcription was then performed in the presence of a template-switching oligonucleotide (5′-GCTAATCATTGCAAGCAGTGGTATCAACGCAGAGTA-CATrGrGrG-3'). Reactions were incubated at 42 °C for 90 min and terminated by heating to 85 °C for 5 min. The resulting cDNA was amplified by PCR using Q5 Hot Start High-Fidelity 2× Master Mix

(NEB, M0494), with 5% of the reverse-transcription product used per reaction. To enhance PCR specificity, nested gene-specific primers located internal to the reverse-transcription primers were used. PCR was performed using a touchdown protocol incorporating progressively lower annealing temperatures, with initial cycles at elevated temperatures (72 °C and 70 °C) to favor target-specific products. Amplified products were resolved on 1.8% agarose gels for visualization or purified using the Zymo DNA Clean and Concentrator kit (D4004) prior to sequencing. Amplicons were sequenced using the MGH CCIB DNA Core complete amplicon sequencing service. Sequencing reads were assembled de novo, and unique 5′ end positions were quantified and displayed in 5′ RACE plots.

## 5′ UTR and transcript annotations

For the analysis of 5′ UTRs, we had to select a representative transcript per gene. We took three independent approaches:

First, we selected the most highly expressed transcript in HeLa cells based on RNA-sequencing. Given the fact that our start site profiling analysis was done in HeLa cells (Ly et al, 2024b), the most highly expressed mRNA isoform can provide the strongest signal to each start codon. To do this, we reanalyzed RNA-sequencing data from control knockout HeLa cells ((Smith et al, 2025), 75 × 75 or 150 bp reads) and G2/M HeLa cells ((Khalizeva et al, 2026); 100 × 100 bp reads). For Kallisto (Bray et al, 2016), we indexed gencode.v47.transcripts.fa (Mudge et al, 2025) and ran Kallisto quant with default settings. We took the average of each transcript abundance, selected the most highly expressed transcript isoform per gene as the representative transcript, an approach we previously used to identify cell-type-specific alternative splicing events (Ly et al, 2024a). If the transcript was not expressed based on the HeLa cell RNA-seq, we selected the longest transcript isoform. The following samples were reanalyzed: GSM9168195, GSM9168199, GSM9380031, GSM9380032, GSM9380039, GSM9380040.

Second, we used MANE select (Morales et al, 2022) as a community standard transcript for clinical reporting. MANE Select is a project that provides a single representative transcript per protein-coding gene that is highly reliable and consistent between the NCBI RefSeq (Pruitt et al, 2007) and Ensembl/GENCODE (Dyer et al, 2025; Mudge et al, 2025) annotations.

Third, we selected the mRNA isoform that had the longest annotated protein-coding sequence from each gene in the Gencode v25 annotations. This transcript choice is frequently used when calculating the translational efficiency of the gene and is an approach that we took in our prior studies (Ly et al, 2024b; Xiang and Bartel, 2021).

The numbers reported in the main text represent analysis using the most highly expressed transcripts from HeLa cells (Smith et al, 2025).

For the assignment of mitochondrial genes, we used Mito-Carta3.0 (Rath et al, 2021).

The sequences of AKR7A2 and ALDH9A1 were retrieved from the NCBI database for conservation analysis. For AKR7A2, the following transcripts were used: NM_003689.4 (human), XM_001163068.7 (chimp), XM_004024785.5 (gorilla), XM_024251104.3 (orangutan), XM_032759127.2 (gibbon), NM_001265698.1 (macaque), and NM_025337.3 (mouse). For ALDH9A1, the following transcripts were used: NM_000696.4 (human),

ENSPTRT00000002946 (chimp), NM_001132693.1 (orangutan), XM_003258802.4 (gibbon), XM_077998992.1 (macaque), and NM_019993.4 (mouse).

## Gene enrichment analysis

For quantitative measurements, such as quantitative IP-MS, we performed Gene Set Enrichment Analysis (GSEA; (Subramanian et al, 2005)) to identify pathways and processes significantly enriched in our IPs. Pre-ranked gene lists were generated based on fold change, and enrichment scores were calculated using default parameters using the fgsea package (1.24.0) in R.

For binned or categorical analyses, such as short vs long 5′ UTR genes, we used Gorilla (Eden et al, 2009) with default settings to identify enriched Gene Ontology (GO) terms. REVIGO (Supek et al, 2011) was used to reduce redundant GO terms.

## Subcellular localization predictions

We used DeepLoc2.1 (Odum et al, 2024), and SignalP (Teufel et al, 2022), webserver with default settings to predict subcellular localization.

## Cap analysis of gene expression (CAGE)

For genome-wide mapping of transcription start sites, total RNA (10 µg per sample) was submitted to an external service provider for Cap Analysis of Gene Expression quality control, library preparation, and sequencing. The raw CAGE-seq reads were mapped to the human genome (GRCh38 Genome Reference Consortium Human Build 38, Dec 2013) using STAR (Dobin et al, 2013) with the default settings except with --sjdbScore 2. The resulting bam file was indexed using samtools index. Aligned reads were processed using bedtools genomecov (v2.30.0) (Quinlan and Hall, 2010) with the parameters -5 -bg -ibam to compute coverage restricted to the 5′ end of each aligned read, generating bedGraph files representing 5′ read end density at single-nucleotide resolution. The resulting bedGraph files were visualized in Integrative Genomics Viewer (IGV, (Thorvaldsdottir et al, 2013)) to examine the distribution of read 5′ ends across genomic loci. Aligned RNA-seq reads were converted to genome-wide coverage tracks using bamCoverage (deepTools v3.5.0, (Ramirez et al, 2014)) with the parameters --binSize 1 --outFileFormat bigwig --normalizeUsing RPKM to generate normalized single-base resolution bigWig files. Enrichment around transcription start sites (based on the Gencode v25 annotations) was quantified using computeMatrix with the parameters -a 1000 -b 1000. Heatmaps were generated from the resulting matrices using plotHeatmap to visualize TSS enrichment.

To call transcription initiation sites for CAGE-Seq data, aligned reads in BAM format were processed to identify transcription start sites (TSSs) based on the 5′ ends of mapped reads, which represent capped RNA molecules. Individual CAGE transcription start sites (CTSS) were extracted from aligned reads by identifying the 5′ position of each read. CTSS positions and their associated read counts were compiled across the genome, generating a map of transcription initiation sites in HeLa cells. To normalize for library size and enable comparison across samples, CTSS expression levels were converted to tags per million (TPM) by dividing raw read counts by the total number of mapped reads and multiplying by

10[6]. Quality filtering was applied to retain only high-confidence TSSs, requiring each CTSS to meet two criteria: (1) TPM ≥ 1, and (2) minimum read count ≥ 5. Filtered CTSS positions were annotated with their nearest gene using custom start codon coordinates derived from the most highly expressed transcript based on RNA seq data in HeLa cells (cite). For each CTSS, we calculated both the signed distance to the annotated start codon. Negative lengths indicate upstream of the start codon, whereas positive lengths indicate downstream of the start codon. All CAGE-seq data processing was performed using custom scripts in R with the GenomicRanges (Lawrence et al, 2013) package from Bioconductor.

## GFP immunoprecipitation

To enrich GFP-tagged annotated or truncated CFD, 5 15-cm plates of cells were pelleted, washed once with cold PBS, once with 50 mM HEPES (pH 7.4), 1 mM EGTA, 1 mM MgCl$_2$, 300 mM KCl, and 10% glycerol, then the pellet was resuspended at a 1:1 ratio in buffer containing 50 mM HEPES (pH 7.4), 1 mM EGTA, 1 mM MgCl$_2$, 300 mM KCl, and 10% glycerol and snap frozen with liquid nitrogen. Cell pellets were thawed following the addition of an equal volume of lysis buffer (75 mM HEPES pH 7.4, 1.5 mM EGTA, 1.5 mM MgCl$_2$, 450 mM KCl, 15% glycerol, 0.075% NP-40) supplemented with EDTA-free protease inhibitors (Roche) and 1 mM PMSF. Cells were lysed by sonication, and insoluble material was removed by centrifugation at 21,000×$g$ for 30 min at 4 °C.

Clarified lysates were incubated with 100 µL Protein A beads conjugated to rabbit anti-GFP antibodies (Cheeseman and Desai, 2005) and rotated end-over-end for 2 h at 4 °C. Beads were washed five times in wash buffer (50 mM HEPES pH 7.4, 1 mM EGTA, 1 mM MgCl$_2$, 300 mM KCl, 10% glycerol, 0.05% NP-40, 1 mM DTT, and protease inhibitors), with each wash performed for 5 min at 4 °C with rotation. Bound proteins were eluted using 3 sequential glycine elutions with 100 µL 100 mM glycine pH 2.6 per elution, and precipitated by the addition of 1/5 volume trichloroacetic acid (TCA) at 4 °C overnight. Precipitates were washed three times with ice-cold acetone and dried by vacuum centrifugation.

## Mass spectrometry

Protein digestion was performed using a modified S-trap workflow (Protifi). Dried TCA pellets were resuspended in S-trap lysis buffer containing 5% SDS and 50 mM TEAB (pH 8.5) and heated to 95 °C for 10 min in the presence of 20 mM DTT. Cysteines were alkylated by treatment with 40 mM iodoacetamide for 30 min at room temperature, followed by acidification to a final concentration of 2.5% phosphoric acid. Six volumes of S-trap binding buffer were added, and samples were loaded onto S-trap micro columns by centrifugation at 4000×$g$. Columns were washed four times with S-trap binding buffer. On-column digestion was carried out overnight at 37 °C using 1 µg trypsin in 50 mM TEAB (pH 8.5). Peptides were sequentially eluted with 20 µL 50 mM TEAB at pH 8.5, then 0.2% formic acid, followed by 50% acetonitrile. The elutions were pooled and quantified using the Pierce Quantitative Fluorescent Peptide Assay (23290) then lyophilized.

Approximately 1.5 µg of digested peptides were resuspended in 50 mM TEAB (pH 8.5) and labeled using TMT10plex reagents at a 20:1 reagent-to-peptide ratio. Labeling reactions were incubated for 1 h at room temperature and quenched by the addition of 0.2% hydroxylamine for 15 min. Labeled samples were pooled on ice, flash-frozen, and lyophilized. Labeled peptides were fractionated using a high-pH reversed-phase peptide fractionation kit (Thermo Fisher Scientific), following the manufacturer's recommendations for TMT-labeled samples. Fractions were pooled (1 + 2, 3 + 4, 5 + 6, and 7 + 8), flash-frozen, and dried.

Dried peptide fractions were resuspended in 0.2% formic acid to a final concentration of 250 ng/µL and analyzed on an Orbitrap Exploris 480 mass spectrometer equipped with a FAIMS Pro interface and coupled to an EASY-nLC system. Peptides were separated on a 25-cm C18 analytical column at a flow rate of 300 nL/min using a multi-step gradient of solvent B. The instrument was operated in positive ion mode with a spray voltage of 1.8 kV and an ion transfer tube temperature of 270 °C. FAIMS analyses were performed using standard resolution settings with alternating compensation voltages across two injections. Full MS scans were acquired in profile mode at 120,000 resolution over an $m/z$ range of 350–1200. The instrument used an automatic determination of maximum fill time, standard automatic gain control (AGC) target, an intensity threshold of $5 \times 10^3$, with precursor selection restricted to charge states +2 to +5, and a dynamic exclusion window of 30 s.

Raw data files were processed using Proteome Discoverer version 2.4 (Thermo Fisher Scientific) to identify proteins and peptides. The Sequest HT (Eng et al, 1994) search engine was employed, using the Homo sapiens protein database (UP000005640) supplemented with EGFP sequences. The search parameters allowed for a maximum of two missed trypsin cleavage sites. Mass tolerances were set to 10 ppm for precursor ions and 0.02 Da for fragment ions. The analysis considered several post-translational modifications: dynamic phosphorylation (+ 79.966 Da on serine, threonine, or tyrosine), dynamic oxidation (+ 15.995 Da on methionine), dynamic acetylation (+ 42.011 Da at the N-terminus), dynamic loss of methionine (− 131.04 Da at the N-terminal methionine), dynamic loss of methionine with acetylation (− 89.03 Da at the N-terminal methionine), static carbamidomethylation (+ 57.021 Da on cysteine), static TMT6plex (+229.163 Da at any N-terminus) and TMT6plex (+229.163 Da on lysine residues). Isotope correction factors for TMT 10plex were applied according to the manufacturer's specifications (Thermo Fisher; product number 90111, lot number VK306786). Peptide identifications were filtered using Percolator to achieve a false discovery rate (FDR) of no more than 0.01. For quantified proteins, a minimum of 5 peptide spectrum matches (PSM) was applied.

## Western blotting

Cells on plates were washed once with 1× PBS. Cells were directly lysed with 1× Laemmli sample buffer (100 mM Tris pH 6.8, 12.5% glycerol (v/v), 1% SDS (w/v), 0.1% bromophenol blue (w/v), 200 mM β-mercaptoethanol). For a confluent 6-well plate, cells were lysed in 200 µL, and scaled equivalently for different amounts of cells. Whole cell extracts were sonicated at 10% amplitude for 5 s using the Branson Digital Sonifier 450 Cell disrupter to sheer genomic DNA, then boiling. Samples were separated by SDS-PAGE and transferred to PVDF or nitrocellulose. Blots were rinsed once

with TBST, then blocked in 5% milk at room temperature for 1 h. Primary antibodies were diluted in 5% milk and incubated with the blot overnight at 4 °C, washed for 5 min with TBST 4×, incubated with secondary antibody in 5% milk for 1 h at room temperature, followed by another four washes with TBST, and rinsed with PBS once. Blots were imaged using an Odyssey Clx machine (LI-COR) and quantified using the Image Studio software (LI-COR).

The following primary antibodies were used: anti-GFP (Roche, 11814460001, AB_390913), anti-alpha tubulin (AbCam, ab52866, AB_869989). The following secondary antibodies were used: IRDye 680RD Goat anti-Rabbit (LI-COR 92668071, AB_10956166), IRDye 680RD Goat anti-Mouse (LI-COR 92668070, AB_10956588), IRDye 800CW Goat anti-Rabbit (LI-COR 92632211, AB_621843), IRDye 800CW Goat anti-Mouse (LI-COR 92632210, AB_621842).

### 5′ UTR ClinVar analysis

To identify ClinVar (Landrum et al, 2018) alleles affecting 5′ UTR length, we downloaded the ClinVar variant summary (December 26, 2025; 4,155,543 GRCh38 variants) and extracted 5′ UTR genomic intervals from GENCODE v47 MANE Select transcripts. We filtered for indels (deletions, insertions, duplications, and insertion+deletion variants) where both start and stop genomic coordinates fell within annotated 5′ UTR regions, yielding 988 variants. We then applied a minimum size threshold of ≥10 bp to focus on variants with meaningful length changes, resulting in 166 variants. Size changes were calculated using genomic coordinates for deletions and HGVS notation parsing for insertions, duplications, and complex indels. Code and analysis are available at https://github.com/cheeseman-lab/5-utr-clinvar-mutations.

### Experimental study design and statistics

No formal statistical method was used to predetermine sample size. No sample size estimate calculation was performed. Sample sizes were chosen according to field standards. No specific randomization was implemented. Blinding was not performed during data collection or analysis. No samples were excluded from analysis. All statistical tests used for each figure are indicated in the corresponding figure legends and were chosen based on the experimental design and data distribution.

## Data availability

The CAGE-Seq data are deposited in GEO with the identifier GSE316181. The mass spectrometry proteomics data have been deposited to the ProteomeXchange Consortium via the PRIDE (Perez-Riverol et al, 2025) partner repository with the dataset identifier PXD073007 and https://doi.org/10.6019/PXD073007. Code and analysis of 5′ UTR ClinVar variants are available at https://github.com/cheeseman-lab/5-utr-clinvar-mutations.

The source data of this paper are collected in the following database record: biostudies:S-SCDT-10_1038-S44319-026-00776-7.

## Peer review information

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

## Acknowledgements

This work was supported by a grant from the NIH (R35GM126930 to I.M.C) and the Chan Zuckerberg Initiative Rare as One Project grant to IMC. JL and YFT are supported in part by the Natural Sciences and Engineering Research Council of Canada. EMS is supported by an American Cancer Society Postdoctoral Fellowship (134255-PF-20-041-01-DMC). MD is supported in part by an NSF GRFP fellowship. Ekaterina Khalizeva is supported by a graduate research fellowship from MIT HEALS. Elizabeth Black is supported by a Damon Runyon post-doc fellowship. We thank the Whitehead Quantitative Proteomics Core for mass spectrometry, the Whitehead Flow Cytometry Core for cell sorting, the Massachusetts General Hospital CCIB DNA core for sequencing, Océane Marescal for technical assistance, and members of the Cheeseman and Bartel labs for helpful discussions. We also thank Dave Bartel, Nicky Whiffin, and Garabet Yeretssian for helpful discussions. This study adheres to the ICMJE recommendations.

## Author contributions

**Jimmy Ly**: Conceptualization; Formal analysis; Funding acquisition; Validation; Investigation; Visualization; Methodology; Writing—original draft; Writing—review and editing. **Eric M Smith**: Funding acquisition; Investigation; Writing—review and editing. **Matteo Di Bernardo**: Funding acquisition; Investigation; Writing—review and editing. **Yi Fei Tao**: Funding acquisition; Investigation; Writing—review and editing. **Elizabeth M Black**: Funding acquisition; Investigation; Writing—review and editing. **Ekaterina Khalizeva**: Funding acquisition; Investigation; Writing—review and editing. **Iain M Cheeseman**: Conceptualization; Supervision; Funding acquisition; Writing—review and editing.

Source data underlying figure panels in this paper may have individual authorship assigned. Where available, figure panel/source data authorship is listed in the following database record: biostudies:S-SCDT-10_1038-S44319-026-00776-7.

## Disclosure and competing interests statement

The authors declare no competing interests.

# Expanded View Figures

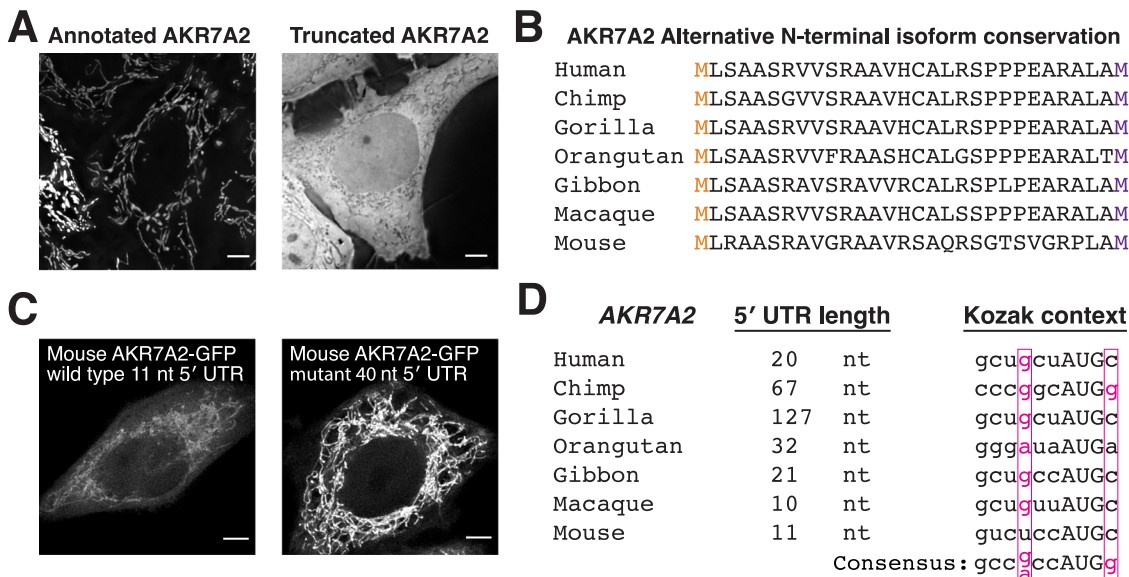

**A** Annotated AKR7A2    Truncated AKR7A2

**B** AKR7A2 Alternative N-terminal isoform conservation

| | |
|---|---|
| Human | MLSAASRVVSRAAVHCALRSPPPEARALAM |
| Chimp | MLSAASGVVSRAAVHCALRSPPPEARALAM |
| Gorilla | MLSAASRVVSRAAVHCALRSPPPEARALAM |
| Orangutan | MLSAASRVVFRAASHCALGSPPPEARALTM |
| Gibbon | MLSAASRAVSRAVVRCALRSPLPEARALAM |
| Macaque | MLSAASRVVSRAAVHCALSSPPPEARALAM |
| Mouse | MLRAASRAVGRAAVRSAQRSGTSVGRPLAM |

**C** Mouse AKR7A2-GFP wild type 11 nt 5' UTR    Mouse AKR7A2-GFP mutant 40 nt 5' UTR

**D**

| AKR7A2 | 5' UTR length | | Kozak context |
|---|---|---|---|
| Human | 20 | nt | gcugcuAUGc |
| Chimp | 67 | nt | cccggcAUGg |
| Gorilla | 127 | nt | gcugcuAUGc |
| Orangutan | 32 | nt | gggauaAUGa |
| Gibbon | 21 | nt | gcugccAUGc |
| Macaque | 10 | nt | gcuguuAUGc |
| Mouse | 11 | nt | gucuccAUGc |
| | | Consensus: | gccgccAUGg |

**Figure EV1. Evolutionary analysis of AKR7A2 alternative isoforms.**

(A) Live imaging of annotated human AKR7A2 and truncated AKR7A2 isoforms. (B) Protein sequence alignment of AKR7A2 from the indicated mammals. The orange M indicates the annotated start site, and the purple M represents the alternative start site. (C) Live imaging of wild-type mouse AKR7A2 and a mutant AKR7A2 mRNA with a longer 5' UTR. Scale bar indicates 5 µm. (D) 5' UTR length and Kozak context of the first start codon of AKR7A2 in the indicated mammals. Source data are available online for this figure.

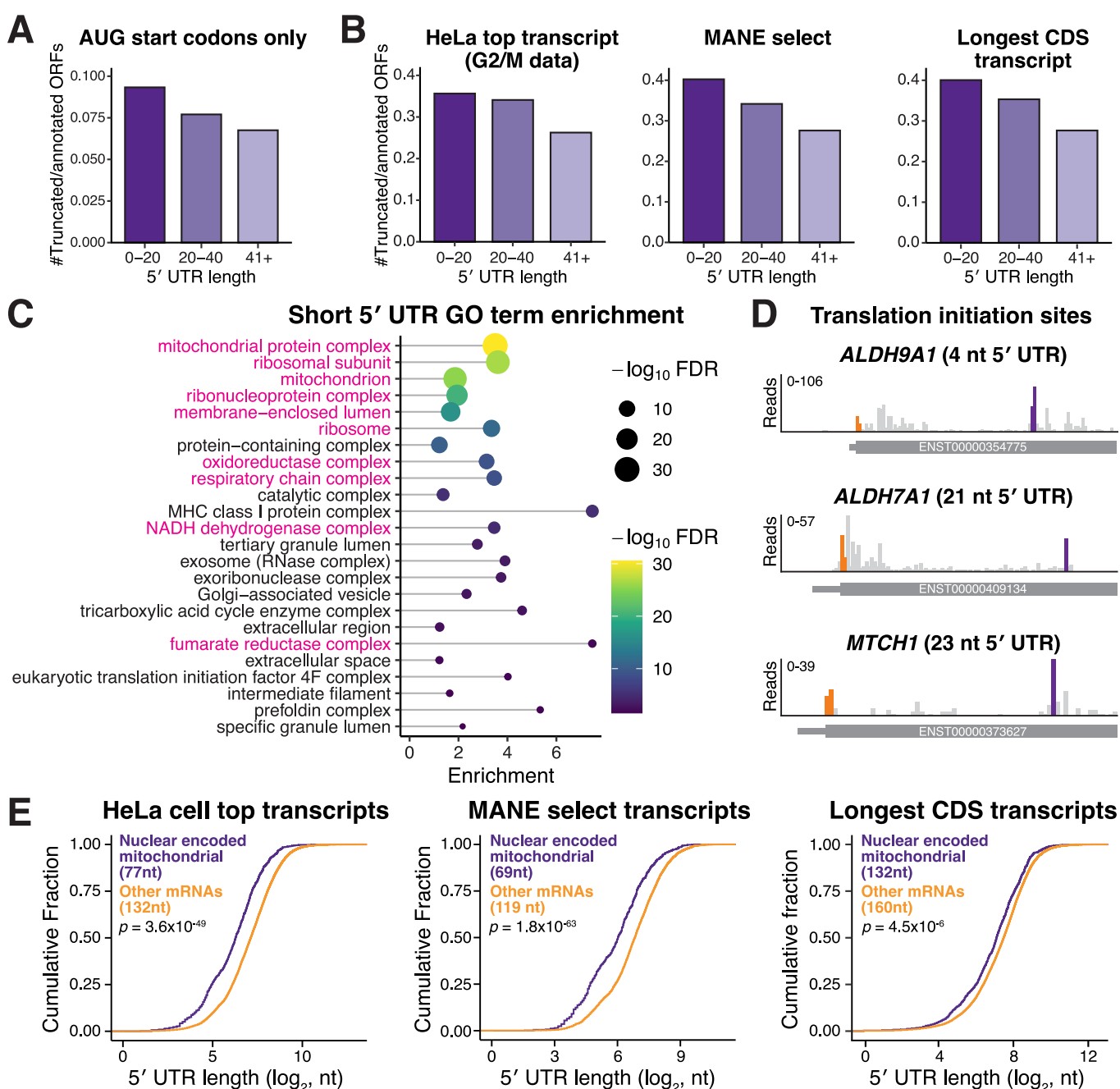

**Figure EV2. Relationship between short 5′ UTRs and N-terminally truncated protein isoforms.**

(A) Same as Fig. 2A except only AUG start codons are included in the analysis. (B) Same as Fig. 2A except the representative transcript isoform from which the 5′ UTR length was calculated was from different datasets ("Methods"). (C) GO term enrichment analysis of short 5′ UTR mRNAs (≤ 40 nt) from the MANE select transcripts (rather than the highest expressed HeLa transcript in Fig. 2B) compared to transcripts with 5′ UTR lengths >40 nt. Pink GO terms indicate mitochondrial-associated terms. (D) Translation initiation site profiling traces around the start codons of indicated mRNAs. (E) CDF plot as described in Fig. 2C, except the representative transcript isoform for each gene was selected by different strategies ("Methods"). Statistics indicate Wilcoxon rank-sum test. Source data are available online for this figure.

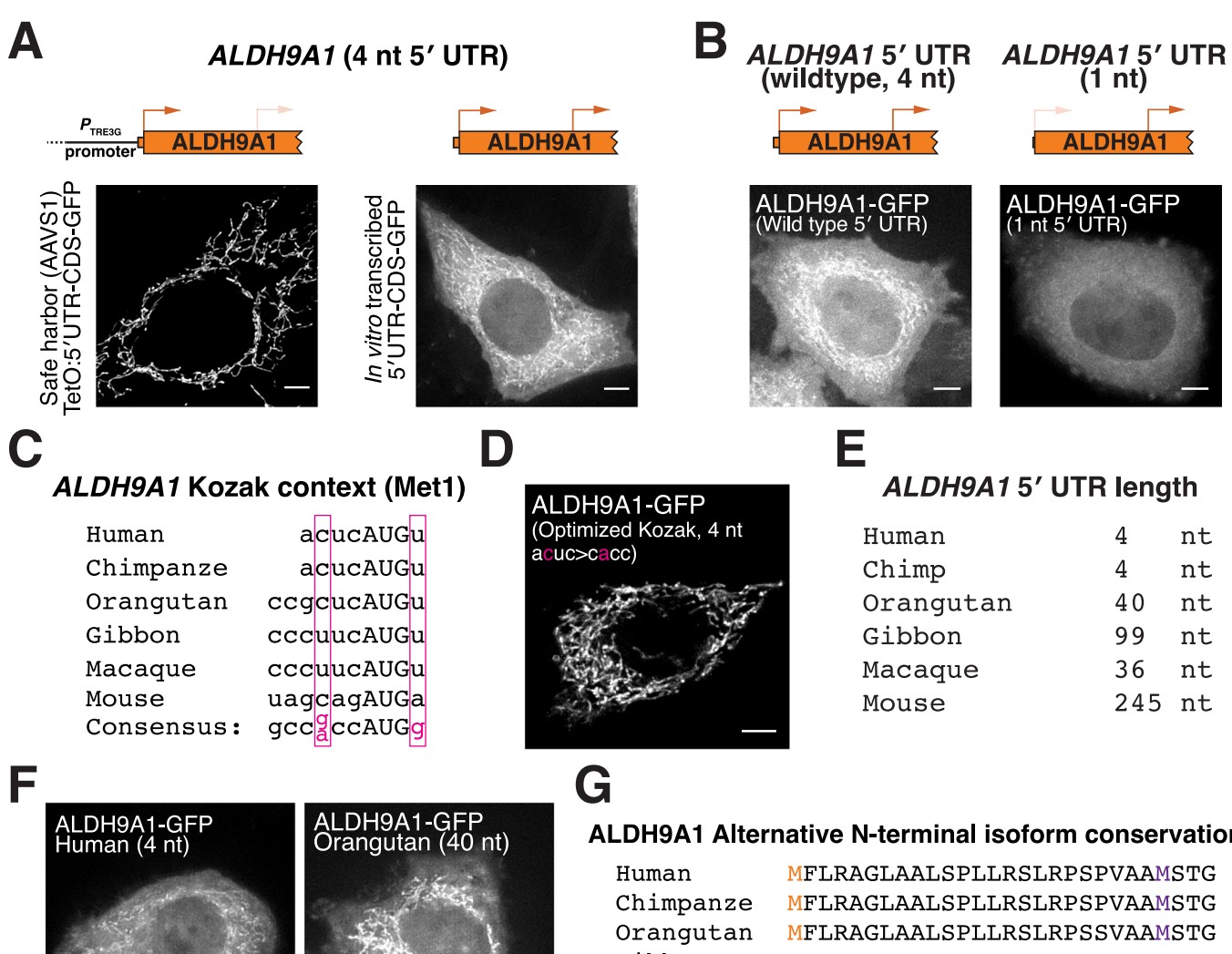

**A** *ALDH9A1* (4 nt 5′ UTR)

Safe harbor (AAVS1) TetO:5′UTR-CDS-GFP

$P_{TRE3G}$ promoter — ALDH9A1

*In vitro* transcribed 5′UTR-CDS-GFP

ALDH9A1

**B** *ALDH9A1* 5′ UTR (wildtype, 4 nt)

ALDH9A1

ALDH9A1-GFP (Wild type 5′ UTR)

*ALDH9A1* 5′ UTR (1 nt)

ALDH9A1

ALDH9A1-GFP (1 nt 5′ UTR)

**C** *ALDH9A1* Kozak context (Met1)

```
Human        acucAUGu
Chimpanze    acucAUGu
Orangutan    ccgcucAUGu
Gibbon       cccuucAUGu
Macaque      cccuucAUGu
Mouse        uagcagAUGa
Consensus:   gccaaccAUGg
```

**D** ALDH9A1-GFP (Optimized Kozak, 4 nt acuc>cacc)

**E** *ALDH9A1* 5′ UTR length

```
Human       4    nt
Chimp       4    nt
Orangutan   40   nt
Gibbon      99   nt
Macaque     36   nt
Mouse       245  nt
```

**F** ALDH9A1-GFP Human (4 nt)

ALDH9A1-GFP Orangutan (40 nt)

**G** **ALDH9A1 Alternative N-terminal isoform conservation**

```
Human       MFLRAGLAALSPLLRSLRPSPVAAMSTG
Chimpanze    MFLRAGLAALSPLLRSLRPSPVAAMSTG
Orangutan   MFLRAGLAALSPLLRSLRPSSVAAMSTG
Gibbon      MFLRAGLALLSPLLRSLRPSPVAAMSTS
Macaque     MFLRAGRAALSPLVRSLQPSPVAAMSTG
Mouse       MILGAVGSVLTSLLRIHRAAAVAAMSTG
```

**Figure EV3.  Additional analysis of ALDH9A1 start codon selection.**

(A) Live-cell imaging of ALDH9A1-GFP produced from dox-inducible promoter or by transfected in vitro-transcribed mRNA. (B) Live-cell imaging of transfected in vitro-transcribed ALDH9A1-GFP with the wild-type (4 nt) or shorter (1 nt) 5′ UTR. (C) Conservation of weak Kozak context around the first AUG in ALDH9A1 across selected mammals. (D) Live-cell imaging of transfected in vitro-transcribed ALDH9A1-GFP with the optimized Kozak context. (E) Analysis of ALDH9A1 5′ UTR lengths across organisms. (F) Live-cell imaging of in vitro-transcribed human or orangutan ALDH9A1-GFP. Scale bar indicates 5 µm. (G) Protein alignment for ALDH9A1 in the indicated organisms. The orange M indicates the annotated start site, and the purple M represents the alternative start site. Source data are available online for this figure.

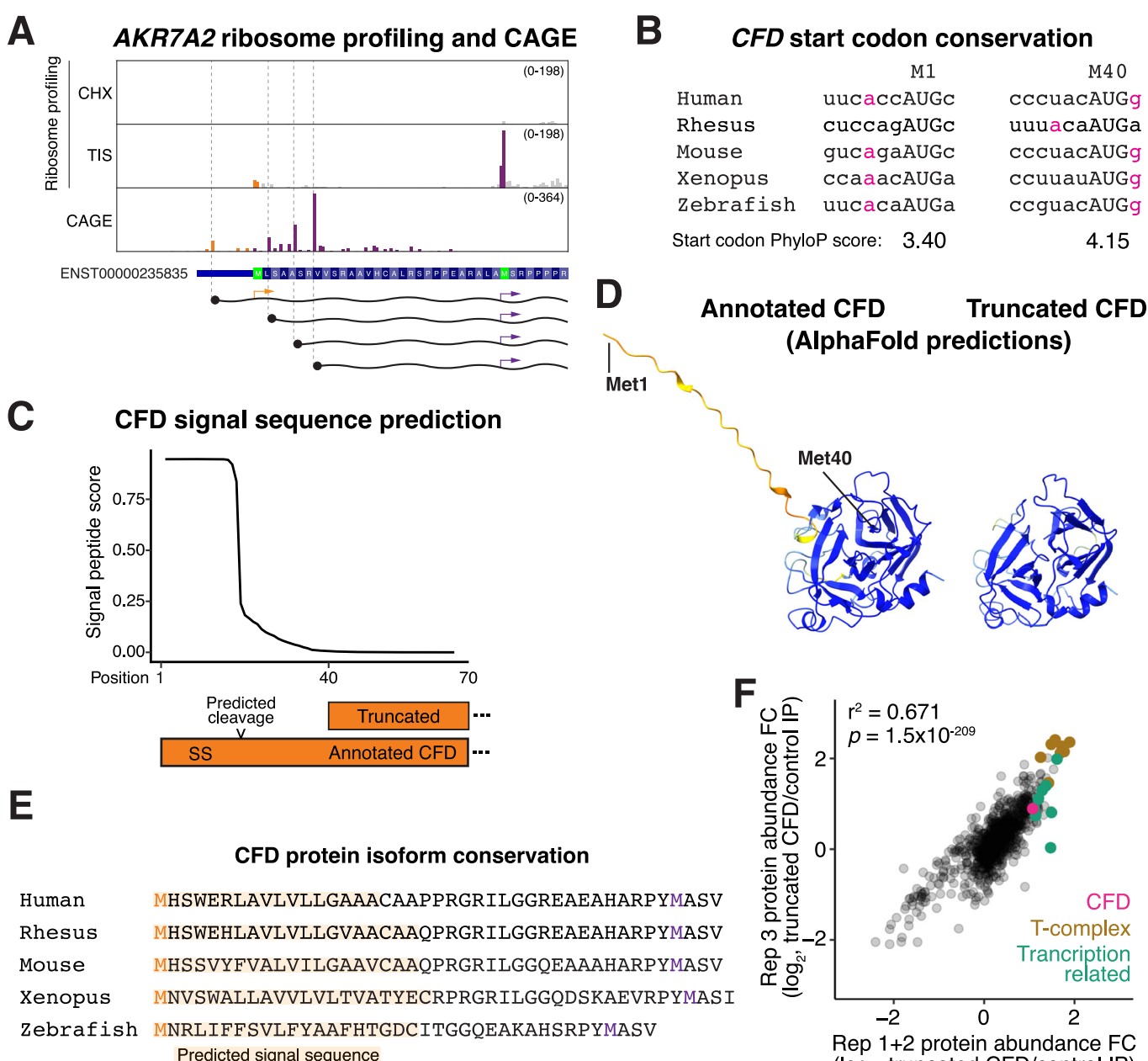

**A** **AKR7A2 ribosome profiling and CAGE**

**B** **CFD start codon conservation**

| | M1 | M40 |
|---|---|---|
| Human | uucaccAUGc | cccuacAUGg |
| Rhesus | cuccagAUGc | uuuacaAUGa |
| Mouse | gucagaAUGc | cccuacAUGg |
| Xenopus | ccaaacAUGa | ccuuauAUGg |
| Zebrafish | uucacaAUGa | ccguacAUGg |

Start codon PhyloP score: 3.40    4.15

**C** **CFD signal sequence prediction**

**D** **Annotated CFD    Truncated CFD (AlphaFold predictions)**

**E** **CFD protein isoform conservation**

| Human | MHSWERLAVLVLLGAAACAAPPRGRILGGREAEAHARPYMASV |
|---|---|
| Rhesus | MHSWEHLAVLVLLGVAACAAQPRGRILGGREAEAHARPYMASV |
| Mouse | MHSSVYFVALVILGAAVCAAQPRGRILGGQEAAAHARPYMASV |
| Xenopus | MNVSWALLAVVLVLTVATYECRPRGRILGGQDSKAEVRPYMASI |
| Zebrafish | MNRLIFFSVLFYAAFHTGDCITGGQEAKAHSRPYMASV |

Predicted signal sequence

**F**

r² = 0.671
p = 1.5x10⁻²⁰⁹

Figure EV4.  Additional analysis of alternative AKR7A2 and CFD isoform.

(A) Read distribution traces from ribosome profiling and CAGE-seq reveals alternative promoter for AKR7A2 that removes the first AUG. (B) Conservation of the CFD start codon across selected organisms. (C) CFD SignalP predicts a strong N-terminal secretion signal. (D) AlphaFold (Abramson et al, 2024) prediction of annotated and truncated CFD isoform. (E) CFD protein sequence alignment. The orange share represents the predicted signal sequence based on SignalP. The orange M indicates the annotated start site, and the purple M represents the alternative start site. (F) Scatterplot of proteins identified in truncated CFD IP-MS across different biological replicates and batches. Statistics indicate $t$ test for Pearson correlation coefficient.

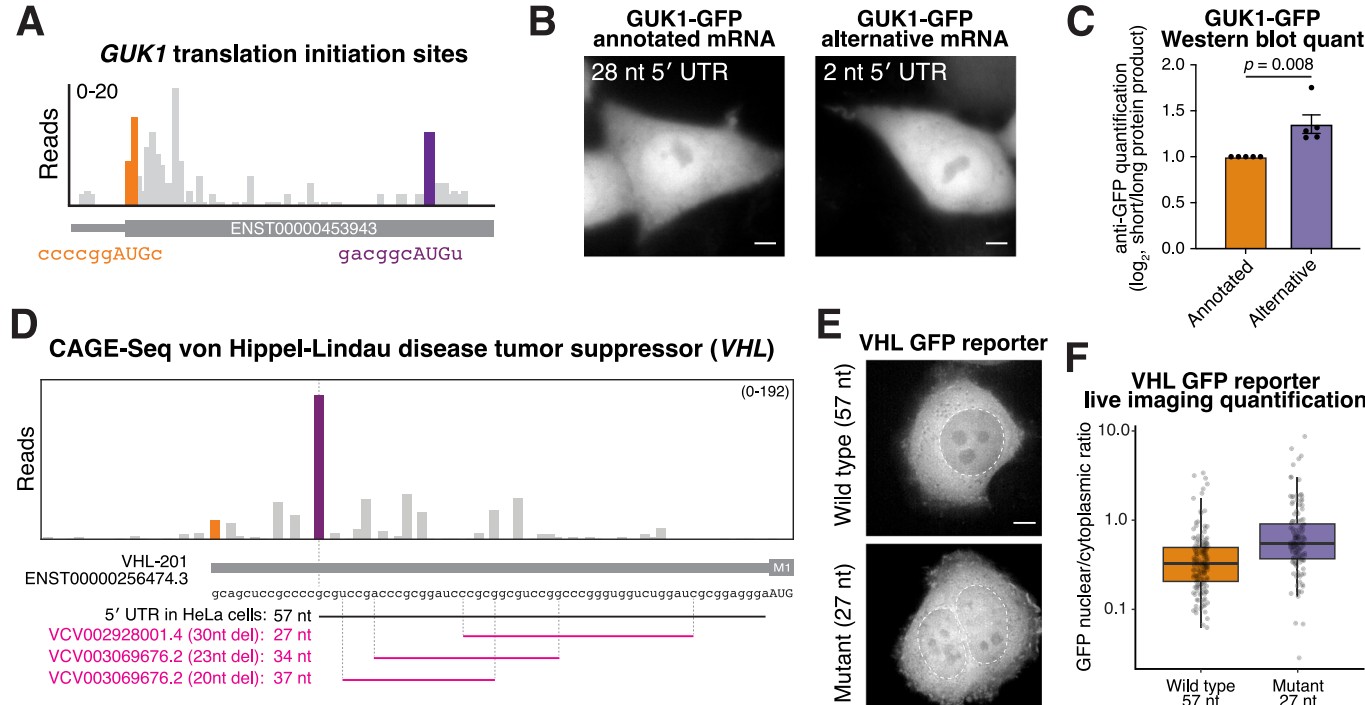

**Figure EV5.** Additional analysis of GUK1 isoforms.

(A) Translation initiation site traces around the start codon for GUK1. (B) Live imaging of transfected annotated or alternative GUK1 mRNA with a C-terminal GFP. Images are not scale equally. Scale bar represents 5 μm. (C) Quantification of Western blot from Fig. 4F. Error bars indicate standard error of the mean. $n = 5$ biological replicates. Statistics indicate unpaired Student's $T$ test. (D) CAGE-seq trace from HeLa cells highlighting that HeLa cells have a shorter 5′ UTR than the annotated isoform. Magenta text and lines represent ClinVar deletions. (E) Representative live imaging of HeLa cells transfected with the indicated in vitro-transcribed reporters. Scale bar indicates 5 μm. (F) Quantification of nuclear vs cytoplasmic ratio of the VHL GFP reporters. Each point represents a single cell. $n = 2$ biological replicates. Boxplot represents 25th, 50th, and 75th percentile, and whiskers indicate 1.5× interquartile range. Source data are available online for this figure.

