## [Peer Review File · EMBO Reports]

5' UTR length shapes alternative N-terminal protein isoforms across cancers and in rare disease

Jimmy Ly, Eric Smith, Matteo Di Bernardo, Yi Fei Tao, Elizabeth Black, Ekaterina Khalizeva, and Iain Cheeseman

Corresponding author(s): Iain Cheeseman (icheese@wi.mit.edu)

Review Timeline:

Submission Date:	29th Jan 26
Editorial Decision:	13th Feb 26
Revision Received:	17th Feb 26
Editorial Decision:	19th Mar 26
Revision Received:	21st Mar 26
Accepted:	31st Mar 26

Editor: Esther Schnapp

Transaction Report:

Dear Iain,

Thank you for the transfer of your manuscript to EMBO reports. We have now received the full set of referee reports that is pasted below.

As you will see, all referees acknowledge that the findings are interesting and that only minor revision are required. I think all referee suggestions are good and should be addressed.

I would thus like to invite you to revise your manuscript with the understanding that the referee concerns must be fully addressed and their suggestions taken on board. Please address all referee concerns in a complete point-by-point response. Acceptance of the manuscript will depend on a positive outcome of a second round of review. It is EMBO reports policy to allow a single round of major revision only and acceptance or rejection of the manuscript will therefore depend on the completeness of your responses included in the next, final version of the manuscript.

We realize that it is difficult to revise to a specific deadline. In the interest of protecting the conceptual advance provided by the work, we recommend a revision within 3 months (16th May 2026). Please discuss the revision progress ahead of this time with the editor if you require more time to complete the revisions.

- 1) A data availability section providing access to data deposited in public databases is missing. If you have not deposited any data, please add a sentence to the data availability section that explains that.
- 2) Your manuscript contains statistics and error bars based on $n=2$. Please use scatter blots in these cases. No statistics should be calculated if $n=2$.

When submitting your revised manuscript, please review the submission guidelines (<https://link.springer.com/journal/44319/submission-guidelines#cms-Revised-submissions>) and carefully review the instructions that follow below. Failure to include requested items will delay the evaluation of your revision.

2) individual production quality figure files as .eps, .tif, .jpg (one file per figure). See <https://media.springernature.com/original/springer-cms/rest/v1/content/27825798/data/v1> for more info on how to prepare your figures.

3) We replaced Supplementary Information with Expanded View (EV) Figures and Tables that are collapsible/expandable online. A maximum of 5 EV Figures can be typeset. EV Figures should be cited as 'Figure EV1, Figure EV2' etc... in the text and their respective legends should be included in the main text after the legends of regular figures.

- For the figures that you do NOT wish to display as Expanded View figures, they should be bundled together with their legends in a single PDF file called *Appendix*, which should start with a short Table of Content. Appendix figures should be referred to in the main text as: "Appendix Figure S1, Appendix Figure S2" etc. See 'Expanded View' section of the submission guidelines.

5) a complete author checklist, which you can download from our author guidelines. Please insert information in the checklist that is also reflected in the manuscript. The completed author checklist will also be part of the RPF.

6) Please note that all corresponding authors are required to supply an ORCID ID for their name upon submission of a revised manuscript (<<https://orcid.org/>>), which can be linked to your profile in the manuscript submission system.

7) Before submitting your revision, primary datasets produced in this study need to be deposited in an appropriate public database. Please remember to provide a reviewer password if the datasets are not yet public. The accession numbers and database should be listed in a formal "Data Availability" section placed after Methods. Please note that the Data Availability Section is restricted to new primary data that are part of this study. * Note - All links should resolve to a page where the data can

be accessed. *

9) Our journal also encourages inclusion of *data citations in the reference list* to directly cite datasets that were re-used and obtained from public databases. Data citations in the article text are distinct from normal bibliographical citations and should directly link to the database records from which the data can be accessed. In the main text, data citations are formatted as follows: "Data ref: Smith et al, 2001" or "Data ref: NCBI Sequence Read Archive PRJNA342805, 2017". In the Reference list, data citations must be labeled with "[DATASET]". A data reference must provide the database name, accession number/identifiers and a resolvable link to the landing page from which the data can be accessed at the end of the reference.

10) Regarding data quantification, the following points must be specified in each figure legend:

- the name of the statistical test used to generate error bars and P values,
- the number (n) of independent experiments (please specify technical or biological replicates) underlying each data point,
- the nature of the bars and error bars (s.d., s.e.m.),
- If the data are obtained from n Program fragment delivered error ``Can't locate object method "less" via package "than" (perhaps you forgot to load "than"?) at //ejpvfs23/sites23b/embo_www/letters/embo_decision_revise_and_review.txt line 54.' 2, use scatter blots showing the individual data points.

11) The journal requires a statement specifying whether or not authors have competing interests (defined as all potential or actual interests that could be perceived to influence the presentation or interpretation of an article). In case of competing interests, this must be specified in your disclosure statement.

12) All Materials and Methods need to be described in the main text using our 'Structured Methods' format, which is required for all research articles. According to this format, the Methods section includes a separate Reagents and Tools Table file (listing key reagents, experimental models, software and relevant equipment and including their sources and relevant identifiers) and a Methods and Protocols section describing the methods using a step-by-step protocol format. The aim is to facilitate adoption of the methodologies across labs. More information on how to adhere to this format as well as a downloadable template (.docx) for the Reagents and Tools Table can be found in our author guidelines.

I look forward to seeing a revised form of your manuscript when it is ready.

Referee #1:

In eukaryotic cells, the small ribosomal subunit binds near the 5' end of an mRNA and slides down (toward 3') scanning the sequence for a start codon. Translation typically initiates at the AUG codon closest to the 5' end of the mRNA. However, some ribosomes slide past a start codon (leaky scan) without initiating. Context nucleotides flanking the start codon - especially at positions -3 and +4 - as well as the levels of the translation factors eIF1 and eIF5 impact the efficiency of start codon selection. Likewise, Marilyn Kozak previously showed that shortening the length of the 5' UTR of a model mRNA increased leaky scanning in vitro. In this paper, the authors provide the first evidence from cells and natural mRNAs supporting the idea that changing the 5' UTR length can change the start site of translation on a natural mRNA. Importantly, as the authors note, this work establishes the value of properly designing expression constructs to produce mRNAs with natural 5' UTRs - and lacking additional sequences that might impact the production of full-length versus N-terminally truncated isoforms.

General comment: the data in the paper support the authors' conclusions and the study provides important new insights on gene expression.

Specific Comments:

1. One main concern regards the live cell imaging data. While the images in Figures 1B, 1D, 1E, 2D, 4C, EV1A, EV1D, EV3A, EV3B, and EV3D clearly show differences in GFP protein localization to cytoplasm versus mitochondria or other subcellular locations, the results are single images. There is no evidence of reproducibility (duplicate or triplicate images in supplement to show that presented image is representative) and/or quantitation of data from multiple experiments.
2. Page 5, line 12 and Figure EV1B: the longer length of the chimp and gorilla 5' UTRs might as the authors note reflect errors in mRNA annotations. However, another possibility is that these species use a different mechanism to generate two proteins from the mRNA. Do chimp and gorilla have different (poorer) context around the first AUG to enhance leaky scanning? When I examined the conservation of sequences using CodeAlignView, I didn't see obvious context changes, but it might be nice to include the context alignment of Met1 in Figure EV1B.
3. Page 2, 7 lines from bottom: shouldn't it read "...occludes a 40-50 nt..."
4. Page 4, line 13: typo in construct name. "UTR-AKR7A2-GFP" not "7A1"
5. Page 5, 4 lines from bottom: I think you mean to cite Fig. 2A, not 2B
6. Page 6, lines 12-17: the authors propose that the weak Kozak context for Met1 of orangutan ALDH9A1 could account for the production of two isoforms of the protein. However, it is noteworthy that orangutan and human have the same context (-3C and +4U). Thus, the authors' data does not support the contention that evolutionary pressure to produce alternate isoforms has led to distinct molecular and regulatory mechanisms in different organisms - both human and orangutan have poor context. If human with a shorter 5'UTR and poor context has more leaky scanning than orangutan with a longer 5'UTR and the same poor context, then perhaps the authors' proposal would be supported.
7. While the CFD example demonstrates an alternate way that changes in transcription can impact protein isoform production, I think it should be emphasized that the different isoforms of CFD are not due to leaky scanning, but rather different mRNAs that either contain or lack the first AUG codon.
8. Page 8, lines 4-9: while the ClinVar mutation in the alternative CFD start codon could suggest that the alternative form is important, it is also possible that proper function of the full-length protein requires an internal methionine at that position and the truncated protein is not functionally significant.
9. Fig EV5C: can a p-value be calculated for the change in the ratio of long to short protein from the two mRNAs?
10. Page 10, line 15: can CRYGD and MYD88 be added to Fig 5B?
11. Page 11, line 2: should cite Fig EV6B-C at the end of this first sentence
12. Page 11, line 7: "underrepresentation" is a single word
13. Page 12, line 2: "isoforms"
14. In the discussion of VHL, it is probably a good idea to mention that Met1 has a poor context (C+4). Somewhat relatedly, in your published start site profiling did you detect translation initiating at Met2 of VHL? I did not see evidence for this on GWIPS (though it could be that Met2 is only used on truncated mRNAs).

15. Page 13, lines 13-14: I think a couple words are missing: "...noncoding mutations can drive pathology not only by altering mRNA stability or overall translational efficiency, but also by rewiring..."

16. Two questions that could possibly be addressed by your data: 1) does Met1 have to have imperfect context to see effect of shorter 5'UTR or is leaky scanning increased on good context as well when the mRNA is shortened? 2) If Met1 has preferred Kozak context does this blunt the impact of a short leader?

Referee #2:

Translational control mechanisms that determine the quantity and identity of synthesized proteins play key roles in gene expression. Here, Ly et al. examine how changes in the length of the 5'UTR affect selection of translation start codons during translation initiation. They demonstrate that very short 5'UTRs ignore the first AUG start codon, regardless of its context, and favor selection of downstream codons. Using CAGE-seq, they define the 5'ends of the mRNAs and identify different length 5'UTRs arise from alternative transcription initiation sites. They also show that mutations in rare genetic diseases alter 5'UTR length of specific mRNAs, which disrupts function of the encoded protein by altering the translation start site.

Overall, the manuscript was concise, well written, and included compelling data that supported their interesting conclusions. I think it will be of broad interest and further highlights the importance of regulated start site selection in gene expression. I have a few minor comments that should be addressed before publication (in no specific order).

1. It is unclear to me how the mRNAs and mechanisms presented here compare to the previously described TISU elements and mRNAs (e.g., PMC3177215 and follow up papers). Is there overlap, or are they distinct? Does the presence or absence of the TISU affect N-terminal tail inclusion/exclusion in the start site ribosome profiling data?

2. "Importantly, a subset of these mRNAs exhibits alternative start codon usage that cannot be readily explained by known translational control mechanisms." I would rephrase this statement because it feels like an unneeded overstatement.

3. In general, the manuscript assumes skipping the first start site in favor of the second occurs due to leaky scanning on short 5'UTRs. While this is one possible mechanism, as the authors likely appreciate, other possibilities also exist. For example, multiple models for how the mRNA template is loaded into the 43S initiation complex have been proposed - e.g., threading versus slotting. In threading, the mRNA 5'-terminus is 'threaded' directly through the mRNA entry channel into the mRNA channel of the 40S subunit and back out the mRNA exit channel, which allows initiator tRNA the opportunity to decode every base in the 5'UTR. In slotting, the 40S subunit latches onto the mRNA to load it into the mRNA channel, which creates a 'blindspot' for start sites, since bases upstream of the ribosomal P site will never encounter the initiator tRNA. So while the effects described in this manuscript could be due to leaky scanning in a threading model, they also are consistent with the slotting model of mRNA loading. A too short 5'UTR will load the first start site into the complex upstream of initiator tRNA. Unless the manuscript contains data that I missed that distinguishes these possibilities, I would suggest the authors present a both models as potential possibilities, especially since the favored mechanism may change depending the mRNA, cell state, etc.

Referee #3:

5' UTR length has previously been associated with translational efficiency, but its role in protein isoform regulation has not been previously reported. In "5' UTR length regulates alternative N-terminal protein isoform production in health and disease" by Ly et al., the authors demonstrate how short 5' UTR length increases leaky ribosome scanning to allow production of alternative N-terminal protein isoform with potentially different cellular localisation patterns and functions. They further provide examples of how alternative transcription initiation can alter 5' UTR length to regulate start codon choices. Moreover, they elucidate the mechanism of a pathologic 5' UTR altering variant in VHL. While the paper shows an interesting and novel role of 5' UTR length, it requires some clarifications, polishing, and increased statistical confidence in analyses before publication.

1. Fig. 2A is never mentioned in the text body. Fig. 2B caption also needs clarification regarding pink GO terms.

2. Figures are mislabeled in VHL 5' UTR reporter section.

3. ALDH9A1 5' UTR length and alternative N-terminal isoform conservation across species are shown in EV3, but only 5' UTR length conservation for AKR7A2 is shown in EV1. The alternative N-terminal isoform conservation for AKR7A2 would be a nice addition.

4. The authors suggest that there is an evolutionary pressure to maintain the production of dual ALDH9A1 translational isoforms, and this process can occur through multiple molecular and regulatory mechanisms. They support this claim using the orangutan 5' UTR as an example. However, EV3 shows that mouse and gibbon have even longer 5' UTR despite having the alternative N-terminal isoform conservation. Adding a potential explanation for the long 5' UTR in discussion would further bolster the author's claim.

5. Some of the experiments are performed in biological duplicates - e.g. IP-MS of truncated CFD and VHL GFP reporter live imaging quantification. These experiments should be performed at least in triplicate to increase statistical confidence of these

findings.

Overall, this paper provides a novel insight on protein isoform selection and highlight the importance of non-coding regions in molecular mechanism and in clinical setting. I think it would be suitable for publication with a minor revision.

1. Does this manuscript report a single key finding? YES/NO

Yes, it reports a novel insight on how 5' UTR length can affect alternative N-terminal protein isoform.

2. Is the reported work of significance (YES), or does it describe a confirmatory finding or one that has already been documented using other methods or in other organisms etc (NO)? YES/NO

Yes

3. Is it of general interest to the molecular biology community? YES/NO

Yes, as the authors mentioned, promoter choice can affect the 5' UTR length and determine the start codon selection, which is relevant when studying genes with alternative N-terminal protein isoforms.

4. Is the single major finding robustly documented using independent lines of experimental evidence (YES), or is it really just a preliminary report requiring significant further data to become convincing, and thus more suited to a longer format article (NO)? YES/NO

Yes

We thank the reviewers for their helpful comment and suggestions. For the revised manuscript, we have added new experiments and analyses to the paper, which we believe provide additional insights on the role of 5' UTR length in alternative isoform selection. These include:

1. **Additional analysis of ALDH9A1 reveals both Kozak context and 5' UTR length are contributors of start codon selection.** The prior version of manuscript suggested that human ALDH9A1 uses a short 5' UTR to promote the production of a truncated isoform. However, this short 5' UTR is not highly conserved across mammals. A reviewer pointed out the possibility that the conserved weak Kozak context may compensate for this increase in 5' UTR length. Indeed, we now show that optimizing the human ALDH9A1 Kozak context eliminates the production of the truncated isoform, suggesting that a poor Kozak context for ALDH9A1 can also promote the production of the alternative isoform.
2. **Further analysis of conservation of short 5' UTR induced alternative isoform selection.** As suggested by the reviewer, we have now performed a more extensive evolutionary analysis of AKR7A2. This analysis revealed that short 5' UTR-induced leaky scanning for AKR7A2 differs across organisms with chimps possibly using alternative transcription instead to promote differential isoform usage.
3. **Clarity on and additional replicates.** We have now provided additional images in the source data. We also performed an additional replicate for the CFD quantitative proteomics, which further support our prior findings.

Below, we have included detailed point-by-point responses to each reviewer comment.

Referee #1:

1. One main concern regards the live cell imaging data. While the images in Figures 1B, 1D, 1E, 2D, 4C, EV1A, EV1D, EV3A, EV3B, and EV3D clearly show differences in GFP protein localization to cytoplasm versus mitochondria or other subcellular locations, the results are single images. There is no evidence of reproducibility (duplicate or triplicate images in supplement to show that presented image is representative) and/or quantitation of data from multiple experiments.

We agree that including additional images is important. We have now provided raw images representing multiple cells in the Source Data.

2. Page 5, line 12 and Figure EV1B: the longer length of the chimp and gorilla 5' UTRs might as the authors note reflect errors in mRNA annotations. However, another possibility is that these species use a different mechanism to generate two proteins from the mRNA. Do chimp and gorilla have different (poorer) context around the first AUG to enhance leaky scanning? When I examined the conservation of sequences using CodeAlignView, I didn't see obvious context changes, but it might be nice to include the context alignment of Met1 in Figure EV1B.

We agree that differences in Kozak context between primates may compensate for the increased 5' UTR length. Consistent with the analysis by this reviewer, we did not observe obvious changes in the Met1 Kozak context (see below). We have now included this information and analysis in the supplement and added a short discussion on the evolution of 5' UTR length.

AKR7A2	5' UTR length		Kozak context
Human	20	nt	gcu ^g gcuAUG ^c
Chimp	67	nt	ccc ^g ggcAUG ^g
Gorilla	127	nt	gcu ^g gcuAUG ^c
Orangutan	32	nt	ggg ^a uaAUG ^a
Gibbon	21	nt	gcu ^g gccAUG ^c
Macaque	10	nt	gcu ^g guuAUG ^c
Mouse	11	nt	gucuccAUG ^c
Consensus :			gcc ^g ccAUG ^g

3. Page 2, 7 lines from bottom: shouldn't it read "...occludes a 40-50 nt..."

Thank you for catching this error.

4. Page 4, line 13: typo in construct name. "UTR-AKR7A2-GFP" not "7A1"

Thank you for catching this error.

5. Page 5, 4 lines from bottom: I think you mean to cite Fig. 2A, not 2B

Thank you for catching this error.

6. Page 6, lines 12-17: the authors propose that the weak Kozak context for Met1 of orangutan ALDH9A1 could account for the production of two isoforms of the protein. However, it is noteworthy that orangutan and human have the same context (-3C and +4U). Thus, the authors' data does not support the contention that evolutionary pressure to produce alternate isoforms has led to distinct molecular and regulatory mechanisms in different organisms - both human and orangutan have poor context. If human with a shorter 5'UTR and poor context has more leaky scanning than orangutan with a longer 5'UTR and the same poor context, then perhaps the authors' proposal would be supported.

Thank you for this suggestion and analyzing these sequences. We agree with this point and we have now rephrased this accordingly. We now propose that humans have two mechanisms to produce alternative ALDH9A1 isoforms—short 5' UTR length and poor Kozak context. Indeed, by optimizing the Met1 Kozak context for ALDH9A1, we found that this favors the usage of the first start site resulting in strictly mitochondrial protein (see below). We have now included this in the Figure EV3D. Because orangutans have a poor Kozak context but not short 5' UTR length, we propose that the lack of short 5' UTR length is tolerated because of the backup weak Kozak context.

7. While the CFD example demonstrates an alternate way that changes in transcription can impact protein isoform production, I think it should be emphasized that the different isoforms of CFD are not due to leaky scanning, but rather different mRNAs that either contain or lack the first AUG codon.

We agree with this point. Our data is consistent with the idea that CFD isoforms do not result from leaky scanning. We have now better emphasized this point in the text.

8. Page 8, lines 4-9: while the ClinVar mutation in the alternative CFD start codon could suggest that the alternative form is important, it is also possible that proper function of the full-length protein requires an internal methionine at that position and the truncated protein is not functionally significant.

We agree that this is a possibility and have modified our text to reflect this.

9. *Fig EV5C: can a p-value be calculated for the change in the ratio of long to short protein from the two mRNAs?*

Thank you for catching this, we have now reported a P-value.

10. *Page 10, line 15: can CRYGD and MYD88 be added to Fig 5B?*

This is a nice addition, thank you for this suggestion.

11. *Page 11, line 2: should cite Fig EV6B-C at the end of this first sentence*

We have now moved the citation to the end of this sentence.

12. *Page 11, line 7: "underrepresentation" is a single word*

Thank you for catching this.

13. *Page 12, line 2: "isoforms"*

Thank you for catching this.

14. *In the discussion of VHL, it is probably a good idea to mention that Met1 has a poor context (C+4). Somewhat relatedly, in your published start site profiling did you detect translation initiating at Met2 of VHL? I did not see evidence for this on GWIPS (though it could be that Met2 is only used on truncated mRNAs).*

We appreciate the suggestion. We have now commented on VHL leaky scanning in the main text when introducing alternative VHL isoforms.

Consistent with the analysis from this reviewer for GWIPS, we did not robustly detect this in our ribosome profiling of HeLa cells. This is in contrast to the multiple published papers on alternative VHL isoforms. We propose that this second start site represents a low usage initiation site that was not robustly detected in ribosome profiling. Indeed, we have observed a similar behavior for Cdc20, with the presence of clear truncated isoforms (cite Tsang) that are not readily detectable using ribosome profiling. We note that it is possible that the clear evidence for this isoform based on western blot assays in published paper represents an increased stability of the truncated isoform such that low amounts of translation can still be observed by western blotting. Overall, given the strong evidence for alternative VHL translation in the literature, we have decided to not go into this controversy.

15. Page 13, lines 13-14: I think a couple words are missing: "...noncoding mutations can drive pathology not only by altering mRNA stability or overall translational efficiency, but also by rewiring..."

We have reworked this sentence.

16. Two questions that could possibly be addressed by your data: 1) does Met1 have to have imperfect context to see effect of shorter 5'UTR or is leaky scanning increased on good context as well when the mRNA is shortened? 2) If Met1 has preferred Kozak context does this blunt the impact of a short leader?

In answer to these questions:

- 1) Anecdotally, short 5' UTRs appear to be associated with imperfect Kozak contexts. In our manuscript, we examined AKR7A2, ALDH9A1, GUK1, and VHL, and in each case the Kozak sequence is suboptimal. However, this rigorous analysis is limited to only four genes, and whether short leaders promote leaky scanning of endogenous mRNAs with strong Kozak contexts remains to be determined.
- 2) By analyzing ALDH9A1 (4 nt 5' UTR with poor context), we found that strengthening the Met1 context blunts the impact of a short 5' UTR. This data is now found in Figure EV3D.

Referee #2:

1. *It is unclear to me how the mRNAs and mechanisms presented here compare to the previously described TISU elements and mRNAs (e.g., PMC3177215 and follow up papers). Is there overlap, or are they distinct? Does the presence or absence of the TISU affect N-terminal tail inclusion/exclusion in the start site ribosome profiling data?*

Our understanding of the literature is that TISU elements promote the usage of the first start codon, despite being cap proximal. We note that, of the 4 mRNAs tested that undergo short 5' UTR induced alternative isoform selection, none of them have a TISU motif. Therefore, it is certainly possible that addition of a TISU motif here would eliminate usage of the downstream start site, but we have not tested this. Whether this represents a generalizable phenomenon is subject to future work. For the revised manuscript we have now commented on TISU elements in the discussion.

2. *"Importantly, a subset of these mRNAs exhibits alternative start codon usage that cannot be readily explained by known translational control mechanisms." I would rephrase this statement because it feels like an unneeded overstatement.*

We agree with this reviewer and we have rephrased this statement.

3. *In general, the manuscript assumes skipping the first start site in favor of the second occurs due to leaky scanning on short 5'UTRs. While this is one possible mechanism, as the authors likely appreciate, other possibilities also exist. For example, multiple models for how the mRNA template is loaded into the 43S initiation complex have been proposed - e.g., threading versus slotting. In threading, the mRNA 5'-terminus is 'threaded' directly through the mRNA entry channel into the mRNA channel of the 40S subunit and back out the mRNA exit channel, which allows initiator tRNA the opportunity to decode every base in the 5'UTR. In slotting, the 40S subunit latches onto the mRNA to load it into the mRNA channel, which creates a 'blindspot' for start sites, since bases upstream of the ribosomal P site will never encounter the initiator tRNA. So while the effects described in this manuscript could be due to leaky scanning in a threading model, they also are consistent with the slotting model of mRNA loading. A too short 5'UTR will load the first start site into the complex upstream of initiator tRNA. Unless the manuscript contains data that I missed that distinguishes these possibilities, I would suggest the authors present a both models as potential possibilities, especially since the favored mechanism may change depending the mRNA, cell state, etc.*

We definitely agree with ribosome slotting as a possible explanation for altered isoform ratios for mRNAs with short 5' UTRs. Our manuscript does not distinguish these possibilities. We believe this would be a fascinating future direction to identify transcripts that directly undergo leaky scanning vs ribosome slotting in vivo and identify cases where this changes across physiology. We have now included a section on these possibilities and some future outlook regarding the roles of ribosome slotting in vivo.

Referee #3:

1. *Fig. 2A is never mentioned in the text body. Fig. 2B caption also needs clarification regarding pink GO terms.*

Thank you for catching these errors, we have now corrected this.

2. *Figures are mislabeled in VHL 5' UTR reporter section.*

Thank you for catching this mistake.

3. *ALDH9A1 5' UTR length and alternative N-terminal isoform conservation across species are shown in EV3, but only 5' UTR length conservation for AKR7A2 is shown in EV1. The alternative N-terminal isoform conservation for AKR7A2 would be a nice addition.*

We agree and have now added this conservation analysis for AKR7A2 to EV1.

4. *The authors suggest that there is an evolutionary pressure to maintain the production of dual ALDH9A1 translational isoforms, and this process can occur through multiple molecular and regulatory mechanisms. They support this claim using the orangutan 5' UTR as an example. However, EV3 shows that mouse and gibbon have even longer 5' UTR despite having the alternative N-terminal isoform conservation. Adding a potential explanation for the long 5' UTR in discussion would further bolster the author's claim.*

We agree and have now added a short section to the discussion related to conservation of these mechanisms. Notably, in chimpanzees, AKR7A2 has two distinct mRNA isoforms (both with long mRNAs) that are predicted to produce the alternative AKR7A2 isoforms. However, different organisms may use distinct mechanisms that remain to be explored.

5. *Some of the experiments are performed in biological duplicates - e.g. IP-MS of truncated CFD and VHL GFP reporter live imaging quantification. These experiments should be performed at least in triplicate to increase statistical confidence of these findings.*

We have conducted an additional replicate of CFD IP-MS and found a strong correlation. However, we note that this IP was conducted as a separate batch of TMT-based proteomics and therefore we cannot pool the 3 samples together. Therefore, we have added a supplemental figure (Fig. EV4F) showing strong correlation with an orthologous biological replicate.

Given that we performed 3 replicates of the western blot assays in both HeLa and HEK293T cells, we are confident in the VHL GFP reporter results. The western blot is the primary assay and directly reads out on the alternative isoforms. The live imaging, although consistent with the western blot, is not a direct readout. Therefore, we have removed this part of the main text but kept this supporting imaging data in with two replicates.

Dear Iain,

Thank you for the submission of your revised manuscript. We have now received the enclosed report from referee 1 and I am happy to say that we can in principle accept your ms now. Only a few editorial requests will need to be addressed before we can proceed with the official acceptance:

- Please add a "Disclosure and Competing Interest Statement" to the ms file.
- The corresponding author needs to be marked on the title page of the ms, and his email needs to be provided too.
- The author credits need to be removed from the ms file. All credits need to be entered during online ms submission.
- The REFERENCE format needs to be corrected, it needs to be alphabetical, not numerical; et al needs to be used after 10 author names. Please use the EMBO reports reference style.
- The following funders are missing in our online submission system as separate entries: American Cancer Society Postdoctoral Fellowship (134255-PF-20-041-01-DMC), NSF GRFP fellowship. All funding information must be in the ms and in our online system.
- Callouts for Fig. 5CD are missing, please add. Fig. 6 does not seem to have any panels labeled, but the following are cited: 6C, 6D-E, please correct.
- The 4 suppl. tables provided (Table S1-Table S4) need to be updated and uploaded as Dataset EV1 - Dataset EV4 in all places (source file names, file types, titles in the system, callouts in the ms and the legends); the legends need to be removed from the ms and each legend should be provided as a separate tab in its Excel file.
- The Materials and Methods section should include a separate Reagents and Tools Table file (listing key reagents, experimental models, software and relevant equipment and including their sources and relevant identifiers) and a Methods and Protocols section in which methods should be described using a step-by-step protocol format with bullet points. More information and a downloadable Table template is available in our guide to authors online: <https://link.springer.com/journal/44319/submission-guidelines>
- Please label the Source Data Excel files (numerical data) with the panels they contain, e.g. Figure 2AB.
- The manuscript sections should be in the following order: Title page - Abstract & Keywords - Introduction - Results - Discussion - Methods - Data Availability - Acknowledgments - Disclosure Statement & Competing Interests - References - Figure Legends - (Main Tables with legends if applicable) - Expanded View Figure Legends.
- Highlights should be removed from the ms file.
- Material and Methods should be renamed to Methods.

* Figure Legends - Comments *

- Please indicate the statistical test used for data analysis in the legends of figures 3F-H; EV4 F
- Please note that the box plots need to be defined in terms of minima, maxima, centre, bounds of box and whiskers, and percentile in the legend of figure EV5 F
- Please note that the error bars are not defined in the legend of figure EV5 C

I would like to suggest to add the word "protein" to the title. Do you agree with

Short 5' UTRs shape alternative N-terminal protein isoforms across cancers and in rare diseases

EMBO press papers are accompanied online by A) a short (1-2 sentences) summary of the findings and their significance, B) 2-3 bullet points highlighting key results and C) a synopsis image that is exactly 550 pixels wide and 200-600 pixels high (the height is variable). The synopsis image should provide a sketch of the major findings, like a graphical abstract. Please note that text needs to be readable at the final size. Please send us this information along with the final manuscript.

Referee #1:

The authors have successfully addressed all the issues I raised. This is a nice story that provides some of the first results from experiments in cells showing that 5' leader length influences start site selection. I strongly recommend acceptance of this important and interesting paper.

All minor editorial requests have been addressed by the authors.

Prof. Iain Cheeseman
Whitehead Institute for Biomedical Research
Massachusetts Institute of Technology
Department of Biology
Suite 401, 455 Main St.
Cambridge, MA 02142
United States

Dear Iain,

I am very pleased to accept your manuscript for publication in the next available issue of EMBO reports. Thank you for your contribution to our journal.

You may qualify for financial assistance for your publication charges - either via a Springer Nature fully open access agreement or an EMBO initiative. Check your eligibility: <https://link.springer.com/journal/44319/how-to-publish-with-us>

>>> Please note that it is EMBO Reports policy for the transcript of the editorial process (containing referee reports and your response letter) to be published as an online supplement to each paper. If you do NOT want this, you will need to inform the Editorial Office via email immediately. More information is available here: <https://link.springer.com/partners/embo-press/editorial-policies#Peer%20review>